# Inter-Agent Relative Representations for Multi-Agent Option Discovery

**Raul D. Steleac**
School of Informatics
University of Edinburgh
raul.steleac@ed.ac.uk

**Mohan Sridharan**
School of Informatics
University of Edinburgh
m.sridharan@ed.ac.uk

**David Abel**
School of Informatics
University of Edinburgh
david.abel@ed.ac.uk

## Abstract

Temporally extended actions improve the ability to explore and plan in single-agent settings. In multi-agent settings, the exponential growth of the joint state space with the number of agents makes coordinated behaviours even more valuable. Yet, this same exponential growth renders the design of multi-agent options particularly challenging. Existing multi-agent option discovery methods often sacrifice coordination by producing loosely coupled or fully independent behaviours. Toward addressing these limitations, we describe a novel approach for multi-agent option discovery. Specifically, we propose a joint-state abstraction that compresses the state space while preserving the information necessary to discover strongly coordinated behaviours. Our approach builds on the inductive bias that synchronisation over agent states provides a natural foundation for coordination in the absence of explicit objectives. We first approximate a fictitious state of maximal alignment with the team, the *Fermat* state, and use it to define a measure of *spreadingness*, capturing team-level misalignment on each individual state dimension. Building on this representation, we then employ a neural graph Laplacian estimator to derive options that capture state synchronisation patterns between agents. We evaluate the resulting options across multiple scenarios in two simulated multi-agent domains, showing that they yield stronger downstream coordination capabilities compared to alternative option discovery methods.

## 1 Introduction

Effective cooperation in complex domains requires agents to distribute the tasks, synchronise actions, and make decisions under partial observability. Humans seeking to cooperate often invent novel strategies for new tasks by adapting cooperation patterns previously learned for related tasks, and reasoning about the requirements of the new tasks (see Appendix A.1 for additional discussion). For example, when learning to play a new ball game, basic cooperation patterns like passing or positioning relative to teammates or opponents surface instinctively and are then adapted to the new setting. This ability to identify and reuse cooperation patterns enables us to bypass relearning basic skills and instead focus on discovering more abstract (high-level) strategies. In this work, we study how to enable AI agents to discover such basic cooperation patterns, and use them to explore and identify more useful cooperation strategies at a higher level than that allowed by primitive actions.

In single-agent reinforcement learning (RL), the *options* framework (Sutton et al., 1999) is a widely used mechanism for formulating temporally extended actions. Options can act as shortcuts between distant regions of the state space during exploration (McGovern & Barto, 2001). Yet, as noted in prior work (Jong et al., 2008), the effectiveness of options is sensitive to many factors, and poorly designed options or a large set of options can hinder learning. This makes *option discovery*, i.e., the automated design of useful options, a challenging problem. Methods based on the eigen-decomposition of the graph Laplacian, such as Eigenoptions (Machado et al., 2017a), have gained traction due to their task-agnostic discovery of options and exploration guarantees (Jinnai et al., 2019a). However, relying on the eigenvectors of the state-transition graph Laplacian leads to an excessive number of options being discovered (twice the state count), a problem that is more pronounced in multi-agent systems because the state space grows exponentially with the number of agents. Moreover, current Laplacian eigenvector approximators are most effective in estimating a

small number of eigenvectors (Wang et al., 2021; Gomez et al., 2024), risking not identifying useful options, particularly those that facilitate exploration at various timescales.

We address option discovery for multi-agent systems through a novel *inter-agent relative state abstraction*. This new state abstraction compresses the joint state space of a group of agents into a compact latent representation centred around the state of maximal alignment among agents, which we call the *Fermat state*. Through this abstraction of the joint state, we drastically reduce the number of discovered options while also focusing the discovery process on how the relationships between the agents change. We then empirically show how this abstraction encourages the emergence of highly coordinated behaviours. Our approach builds on the intuition that in the absence of an explicit objective, synchronisation over state features represents a natural basis for coordination. Returning to the ball game example, both passing and positioning skills can be understood as forms of multi-agent state synchronisation; they determine who holds the ball and how the players align relative to one another along each spatial dimension. This intuition is consistent with insights into position alignment in animal collective movement (Herbert-Read, 2016) and emergent coordination in human psychology (Knoblich et al., 2011). The key contributions of our paper are:

- A novel abstract joint state representation that maps the relationships between the agents to a multi-dimensional *N-metric space*;
- The use of the abstract state representation, and its ability to compress and reorient eigenoptions toward inter-agent relations, for the discovery of highly coordinated joint options; and
- Adapting the MacDec-POMDP framework (Amato et al., 2019) to support joint-option execution.

We illustrate and experimentally evaluate the capabilities of our approach in two benchmark multi-agent collaboration domains: Level-Based Foraging (Papoudakis et al., 2021) and Overcooked (Ruhdorfer et al., 2024). We demonstrate the benefits of incorporating joint options over baselines that do not include such options, and show that the coordination behaviours discovered using our proposed state abstraction lead to better performance than the behaviours obtained using other multi-agent option discovery methods.

## 2 BACKGROUND

We begin by describing the basic concepts of Dec-POMDPs, the options framework, and $n$-metrics.

### 2.1 DECENTRALISED PARTIALLY-OBSERVABLE MARKOV DECISION PROCESSES

A Dec-POMDP (Bernstein et al., 2009) is defined as a tuple $\langle \mathcal{I}, \mathcal{S}, \{\mathcal{A}^i\}, \mathcal{T}, \mathcal{R}, \{\Omega^i\}, \mathcal{O}, \gamma \rangle$, where $\mathcal{I} = \{1, \ldots, N\}$ is a finite set of agents' indices, $\mathcal{S}$ the state space and $\mathcal{A} = \mathcal{A}^1 \times \cdots \times \mathcal{A}^N$ the joint action space. At every time-step $t$, agent $i$ receives its local observation $o_t^i \in \Omega^i$, where $\Omega = \Omega^1 \times \cdots \times \Omega^N$ is the joint set of observations that was generated according to the observation function $\mathcal{O} : \mathcal{S} \times \mathcal{A} \times \Omega \to [0, 1]$. Agent $i$ then selects an action $a_t^i \in \mathcal{A}^i$ according to a policy $\pi^i(a_t^i | h_t^i)$, that is conditioned on the history of its local observations and actions $h_t^i = (o_1^i, a_1^i, \ldots, o_{t-1}^i, a_{t-1}^i, o_t^i)$. Given the joint set of actions at $t$, $a_t = \{a_t^1, \ldots, a_t^N\}$, the environment transitions to a new state $s_{t+1} \in \mathcal{S}$ according to the state transition function $\mathcal{T} : \mathcal{S} \times \mathcal{A} \times \mathcal{S} \to [0, 1]$ and induces a global reward received by all agents $r_t = \mathcal{R}(s, a)$, where $\mathcal{R} : \mathcal{S} \times \mathcal{A} \to \mathbb{R}$. The goal is to learn a joint policy $\pi = (\pi^1, \ldots, \pi^N)$ that maximises the expected cumulative discounted return $G = \sum_{t=1}^{T} \gamma^{t-1} r_t$.

Amato et al. (2019) extended Dec-POMDPs by integrating single-agent macro-actions (options) within an asynchronous execution scheme. They defined a MacDec-POMDP as the tuple $\langle \mathcal{I}, \mathcal{S}, \mathcal{A}, \{\mathcal{M}^i\}, \mathcal{T}, \mathcal{R}, \{\mathcal{Z}^i\}, \{\Omega^i\}, \{\zeta^i\}, O \rangle$, where $\langle \mathcal{I}, \mathcal{S}, \mathcal{A}, \mathcal{T}, \mathcal{R}, \{\Omega^i\}, O \rangle$ are the same as in a Dec-POMDP. The primitive action set of each agent $i \in \mathcal{I}$, $\mathcal{A}^i$, is replaced with a finite set of macro-actions $\mathcal{M}^i$, and $\mathcal{M} = \mathcal{M}^1 \times \cdots \times \mathcal{M}^N$ is the joint macro-action set. They also introduced a joint set of macro-observation $\zeta = \zeta^1 \times \cdots \times \zeta^N$, with $\mathcal{Z}^i : \mathcal{M}^i \times \mathcal{S} \times \zeta^i \to [0, 1]$ specifying the macro-observation probability function for each agent $i$. To formalise macro-actions, separate histories are maintained for the two execution levels: $H_A^i$ is the *primitive* action–observation history, and $H_M^i$ is the *macro-action–macro-observation* history. The joint primitive history is $H_A = (H_A^1, \ldots, H_A^N)$, and the joint macro history is $H_M = (H_M^1, \ldots, H_M^N)$.

## 2.2 OPTIONS

Sutton et al. (1999) define an option as a tuple $w = \langle I_w, \pi_w, \beta_w \rangle$, where $I_w \subseteq S$ is the initiation set where the option can be invoked, $\pi_w$ is the option policy, and $\beta_w : S \to [0, 1]$ is the termination condition. If these components depend only on the current state, the option is referred to as a *Markov option*. This notion is generalised to *semi-Markov options*, in which the policy and termination condition may depend on additional information (e.g., state–action–reward histories), or termination may be triggered by external factors such as a fixed $k$-step horizon.

Eigenoption discovery (Machado et al., 2017b) estimates the eigenvectors of the combinatorial graph Laplacian corresponding to a state–transition graph, typically via random walks. The eigenvectors of the graph Laplacian, $L = D - A$ (where D and A are the degree and adjacency matrices, respectively), captures long-term temporal relationships between states and the overall geometry of an MDP (Mahadevan & Maggioni, 2007). Given an eigenvector $\mathbf{e}$, the intrinsic reward function for transitioning from discrete state $s$ to $s'$ can be computed as $r_{\mathbf{e}}(s, s') = \mathbf{e}(s') - \mathbf{e}(s)$. Subsequent work (Wu et al., 2018; Jinnai et al., 2020; Wang et al., 2021) extends this framework to non-tabular domains by approximating the eigenvectors through neural networks trained to minimise objectives derived from *graph-drawing theory* (Koren, 2005). The ALLO method introduced by Gomez et al. (2024) further improves robustness to hyperparameters and eigenvector rotations.

## 2.3 $n$-METRICS AND $n$-DISTANCES

**$n$-metric**. Given a set $X$ and an integer $n \geq 2$, an $n$-(hemi)metric (Deza & Rosenberg, 2000; Deza & Deza, 2009) is a function $d : X^n \to \mathbb{R}$, that respects the following conditions:

(M1) (*Non-negativity*) $d(x_1, \ldots, x_n) \geq 0$ for all $x_1, \ldots, x_n \in X$.
(M2) (*Total symmetry*) $d(x_1, \ldots, x_n) = d(x_{\pi(1)}, \ldots, x_{\pi(n)})$, for all $x_1, \ldots, x_n \in X$ and for any permutation $\pi$ of $\{1, \ldots, n\}$.
(M3) (*Definiteness*) $d(x_1, \ldots, x_n) = 0$, if and only if $x_1, \ldots, x_n$ are not pairwise distinct.
(M4) (*Simplex inequality*) $d(x_1, \ldots, x_n) \leq \sum_{i=1}^{n} d(x_1, \ldots, x_n)_i^z$, for all $x_1, \ldots, x_n, z \in X$.

This definition uses $n$ to replace $m + 1$ in the original definition of an *m-hemimetric* (Deza & Deza, 2009) and uses $d(x_1, \ldots, x_n)_i^z$ to denote the function evaluated on $n$ elements where the $i$-th element $x_i$ is replaced by $z$ drawn from the same set $X$.

**$n$-distance.** $n$-distances (Martín & Mayor, 2011; Kiss et al., 2018) relax (M3) by setting $d(x_1, \ldots, x_n) = 0$ only when $x_1 = \ldots = x_n$, providing a direct way of comparing the *dissimilarity* or *separateness* for sets with more than two elements.

**Fermat $n$-distance.** Given a metric space $(X, d)$ and an integer $n \geq 2$, a *Fermat set $F_x$* for a list of $n$ elements $(x_1, \ldots, x_n)$ is a set that minimises the sum of distances to each element in the list (Boltianski et al., 1998):

$$F_X = \left\{ x \in X : \sum_{i=1}^{n} d(x_i, x) \leq \sum_{i=1}^{n} d(x_i, x'), \forall x' \in X \right\}. \tag{1}$$

The elements of $F_x$ are then named the *Fermat points* for the respective list. Based on these definitions, Kiss et al. (2018) introduce Fermat $n$-distances as functions $d_F : X^n \to \mathbb{R}$ of the form:

$$d_F(x_1, \ldots, x_n) = \min_{x \in X} \sum_{i=1}^{n} d(x_i, x). \tag{2}$$

## 3 MULTI-AGENT OPTION-DISCOVERY

We next describe our framework for multi-agent option discovery, starting with the method for approximating the *spread* of a group of agents through $n$-distance estimation.

## 3.1 $n$-DISTANCES FOR MULTI-AGENT DISSIMILARITY ESTIMATION

In the absence of a reward signal, one key strategy for coordination among a group of agents is through the alignment of their states. An important step to generate such collaborative behaviours

is to define a measure of *spreadingness* for the group of agents at any given time. We begin by introducing a notion of distance between the state of two agents. To this end, we assume that the joint state-space, $\mathcal{S}$, can be factored into $N$ single-agent state spaces $\mathcal{S}^i$, with $N = |\mathcal{I}|$ and $i \in \mathcal{I}$, by ignoring the presence of others and including information corresponding solely to each individual agent [1]. We denote $s^i \in \mathcal{S}^i$ to be a single agent-state, and $d(s^i, s^j)$ as a distance metric capable of comparing the similarity between the state of two agents, $i, j \in \mathcal{I}$. For simplicity, we focus our notation and definitions on homogeneous agent state spaces, i.e., $\mathcal{S}^* = \mathcal{S}^1 = \ldots = \mathcal{S}^N$. However, Appendix A.8 describes how this model can be extended to heterogeneous settings, and presents empirical evaluation in two toy scenarios.

Inspired by *Fermat $n$-distances* (Equation 2), we define an $n$-distance metric for a group of agents.

**Definition 1.** *For a metric space $(\mathcal{S}^*, d)$, where $\mathcal{S}^*$ is a single-agent state space, $d$ is a state distance metric and $N \geq 2$, we define the **Fermat inter-agent state distance** as a map $d_{\mathcal{F}} : \mathcal{S} \to \mathbb{R}$ such that:*

$$d_{\mathcal{F}}(s^1, \ldots, s^N) = \min_{s \in \mathcal{S}^*} \sum_{i=1}^{N} d(s^i, s). \tag{3}$$

Computing the minimization operation from Definition 1 becomes intractable in large or continuous state spaces. To alleviate this problem, we propose approximating the *Fermat* state, the state of minimum summed distance to each agent, through a parameterised function, $\phi : \mathcal{S} \to \mathcal{S}^*$, which we call the Fermat encoder and train by minimizing the following objective:

$$\mathcal{L}_{\mathcal{F}}(\phi, d) = \mathbb{E}_{\tau \sim \rho_{\pi}} \left[ \frac{1}{N} \sum_{i=1}^{N} d(s_t^i, \phi(s_t))^2 \right]. \tag{4}$$

where $\tau \triangleq (s_0, \ldots, s_{T-1}) \sim \rho_{\pi}$ is a history of states under the trajectory distribution $\rho_{\pi}$ of joint policy $\pi$, $T$ is the time horizon of an episode, $s_t$ is the joint state, and $s_t^i$ is the $i$th element of the factorised joint state, corresponding to agent $i$. This objective depends on a pre-defined state distance metric $d$. While any valid state distance metric would suffice, we employ *temporal* distances due to their invariance to feature semantics and close alignment with environmental dynamics (see Section 5). Temporal distances are typically formulated as quasimetrics that are obtained by relaxing the symmetry requirement to account for the *arrow of time* (e.g., ascending a mountain takes more time than descending it). Although a quasimetric function can be symmetrised, e.g. $d_m(x, y) = d_q(x, y) + d_q(y, x)$, such transformations remove a key advantage of *temporal distances* and reduce the expressivity of the resulting measure. Thus, we enforce a consistent input order by fixing the Fermat state as the *second* input in Equation 4, yielding a directed function that can be interpreted as the expected number of steps needed for the agents to achieve full alignment.

We adopt the *successor* distances method (Myers et al., 2024) for approximating temporal distances and denote the parameterised state distance as $d_{\theta} : \mathcal{S}^* \times \mathcal{S}^* \to \mathbb{R}$. The Fermat encoder $\phi$ and the distance approximator $d_{\theta}$ are trained concomitantly by enforcing a *stop-gradient* operator on the distance estimator's parameters ($\theta$) when integrating it in the Fermat encoder objective (Equation 4).

### 3.2 MULTI-AGENT OPTION DISCOVERY ON RELATIVE STATES

Eigenoption discovery, introduced in Section 2.2, consists of two main steps: (i) estimating the state-transition graph and (ii) performing the eigen-decomposition of the graph Laplacian to generate a set of eigenvectors for option training. An important observation is that this process is completely dependent on the state representation used to construct the transition graph. We propose to *intentionally leverage* this observation by embedding the joint state space into an inter-agent relative representation prior to performing the graph Laplacian eigen-decomposition.

Intuitively, one could replace each joint state in the transition graph with the corresponding $n$-distance estimation. However, such a compression may limit the behavioural expressivity captured in the eigenvectors. Using a singular scalar value to describe the dissimilarity on all feature dimensions can obscure their individual effect. For instance, two agents reported as being $k$ units apart may differ primarily along one dimension defining the space, or along multiple dimensions, but the

---

[1]Single agent states can contain general environmental information. Because it is shared by all agents, it will be naturally ignored by the state distance measure.

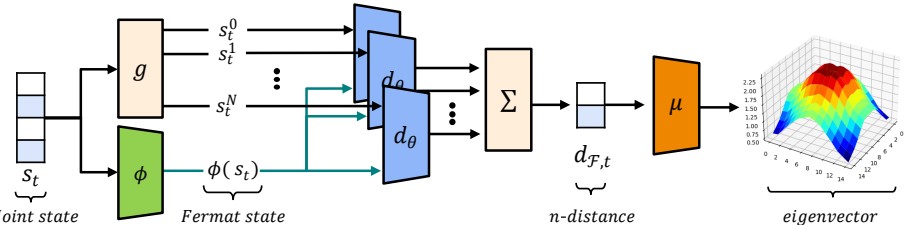

Figure 1: Option discovery on *inter-agent* relative representations for a state factorisation function $g$, Fermat encoder $\phi$, state distance encoder $d_\theta$ and a graph Laplacian eigenvector approximator $\mu$.

scalar provides no insight into which dimension drives the misalignment. Furthermore, mapping joint states to scalar values has drastic effects on the topology of the state-transition graph, further limiting the diversity of the alignment behaviour discovered. To address this issue, we compute the $n$-distance along each feature dimension of the single-agent state space and encode the nodes in the state-transition graph as their concatenation. The distance module is thus modified to predict $F$ outputs $d^F : \mathcal{S}^* \times \mathcal{S}^* \to \mathbb{R}^F$, where $F = \dim(\mathcal{S}^*)$. We then use a linear projection layer to approximate the overall state-distance when training $d_\theta^F$. Figure 1 provides an overview of our joint option discovery framework, which uses multi-dimensional $n$-distances as state representations.

In practice, however, this unconstrained decomposition can lead to degenerate solutions, as $d_\theta^F$ can output identical distance estimates on all dimensions or revert to only using one of the output dimensions. To mitigate this issue, we impose a Mutual Information (MI) objective that encourages each state feature to dominate the information flow leading to its corresponding distance prediction. Formally, let $S^{i,j} = (S^i, S^j)$ be the random pair of single-agent states corresponding to agents $i, j \in \mathcal{I}$, with $i \neq j$, obtained by factorizing states of joint trajectories from $\rho_\pi$. For each feature index $f \in \{1, \ldots, F\}$, we denote $S_f^{i,j} = (S_f^i, S_f^j)$ as the pair of their $f$-th feature dimension. Given that $d^F : \mathcal{S}^* \times \mathcal{S}^* \to \mathbb{R}^{\mathbb{F}}$ is a deterministic map, we define a random vector $Z^{i,j} = d^F(S^i, S^j) \in R^F$ and its $f$-th component $Z_f^{i,j}$. We require that the MI between each feature distance $Z_f^{i,j}$ and the rest of the state-feature pairs $S_{-f}^{i,j}$, i.e. $\mathbb{I}(S_{-f}^{i,j}, Z_f^{i,j})$, does not exceed the MI between the feature pairs themselves, $\mathbb{I}(S_{-f}^{i,j}, S_f^{i,j})$. This implies that each distance prediction should not carry more information about the value of other features than is already captured in the correlations between the inherent features. We next establish that through simple manipulations of the chain rule of information, we can upper bound the information between $Z_f^{i,j}$ and the rest of the state features $S_{-f}^{i,j}$ by the information already explained by feature $S_f^{i,j}$ about $S_{-f}^{i,j}$ plus a residual conditional term.

**Proposition 1.** *For any two agent indexes $i, j \in \mathcal{I}$, with $i \neq j$, and feature index $f \in \{1, \ldots, F\}$:*

$$\mathbb{I}(S_{-f}^{i,j}; Z_f^{i,j}) \leq \mathbb{I}(S_{-f}^{i,j}; S_f^{i,j}) + \mathbb{I}(S_{-f}^{i,j}; Z_f^{i,j} \mid S_f^{i,j}). \tag{5}$$

A proof of this proposition is in Appendix A.2. To reduce information overflow in the distance predictions, we thus minimise the following Conditional Mutual Information (CMI), measuring the excess information that $Z_f^{i,j}$ conveys about $S_{-f}^{i,j}$ beyond what is already captured in $S_f^{i,j}$:

$$\mathbb{I}(S_{-f}^{i,j}; Z_f^{i,j} \mid S_f^{i,j}) = D_{\mathrm{KL}}\left[p(S_{-f}^{i,j}; S_f^{i,j}; Z_f^{i,j}) \,\middle\|\, p(S_{-f}^{i,j} \mid S_f^{i,j}) p(Z_f^{i,j} \mid S_f^{i,j})\right] \tag{6}$$

**Minimizing CMI.** We adopt the approach of Dunion et al. (2023) and introduce a discriminator network $D_\psi$ trained to directly detect the information excess. Specifically, the discriminator is trained to distinguishing between real triplets $\{s_{-f,t}^{i,j}; s_{f,t}^{i,j}; z_{f,t}^{i,j}\}$ obtained by pairing factorised single-agent states from a history of joint states $\tau \triangleq (s_0, \ldots, s_T) \sim \rho_\pi$, and fake triplets obtained by permuting $s_{-f,t}^{i,j}$ while keeping $\{s_{f,t}^{i,j}; z_{f,t}^{i,j}\}$ fixed. As in Dunion et al. (2023), we use the discriminator's predictions to define a disentanglement penalty when training $d_\theta^F$. We defer a description of the CMI minimization algorithm to Appendix A.3. A visual comparison of scalar and the multi-dimensional $n$-distances is in Appendix A.5.

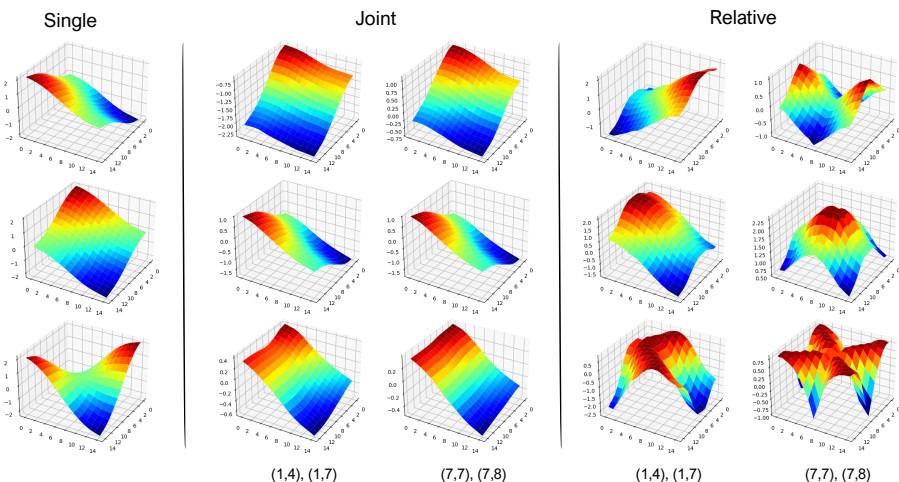

Figure 2: The first three non-trivial eigenvectors of the graph Laplacian for an 15x15 grid environment with three agents, as the only entities in the grid, under varying state representations: single agent state spaces (left), raw joint state spaces (centre) and inter-agent relative state representations (right). For visibility, we fix the position of two agents for the multi-agent scenarios (at [(1,4), (1,7)] and [(7,7), (7,8)]), and display the values when varying the position of the remaining agent.

We follow the representation-driven option discovery (ROD) cycle from Machado et al. (2023) [2] to generate the set of options, but precede the eigenvector estimation by first representing joint states as their disentangled multi-dimensional $n$-distance representation. Both the $n$-distance encoder training and joint option discovery are done prior to the generic training. By converting raw joint states into relative representations, we produce options that reflect a range of complex multi-agent alignment behaviours, enabling agents to synchronise along various subsets of their state features. Figure 2 (right) illustrates the resulting eigenvectors in a grid-world setting with coordinate-based states; Appendix A.6 extends this analysis to prior multi-agent eigenoption methods (Chen et al., 2022). For ease of understanding, we fix the positions of $N - 1$ agents and visualise the eigenvectors from the remaining agent's perspective. We emphasise that these relative eigenvectors are highly responsive to changes in the joint state due to the re-centring effect around the *Fermat* state, a property that is much less evident in the eigenvectors discovered on raw joint states. In this simple domain, the first eigenvector aligns agents along one coordinate axis, while its negation (also a valid eigenvector) aligns them along the other. The subsequent eigenvector promotes alignment across both axes simultaneously, followed by eigenvectors that capture more complex synchronisation patterns.

Following the procedure introduced in Section 2.2, we use these eigenvectors as intrinsic reward signals for option-policy training. Figure 3 shows a visualisation of the multi-agent options trained to follow the positive and negative versions of the first two eigenvectors for a team of four agents. With each grid, we report the feature-wise distance values for the corresponding final states, highlighting the distinct alignment patterns achieved by the learned option policies. Furthermore, the first two eigenvectors induce the same behaviours for four agents as they did for three in Figure 2, illustrating the consistency of the alignment patterns with respect to team size.

## 3.3 ADDING JOINT OPTIONS TO DEC-POMPDS

We adapt the MacDec-POMDP framework (Amato et al., 2019) described in Section 2.1 to support multi-agent macro-actions (*joint options*). To ensure the correct selection, execution, and termination of joint options in decentralised settings, we impose the following two modeling assumptions:

**Assumption 1.** *There is an information-sharing mechanism between all agents.*

This assumption yields a more permissive model than the standard MacDec-POMDP framework. However, it still withholds access to the true underlying state. The information sharing mechanism

---

[2]We use the ALLO method from Gomez et al. (2024) to approximate the eigenvectors of the graph Laplacian and follow the original *eigenoptions* approach from Machado et al. (2017b), where we use a single ROD cycle to generate the whole set of eigenvectors. A motivation for these design choices is in Appendix A.9.



Figure 3: Policy roll-outs visualisation of the first four relative options in the 15×15 grid environment with four agents. Arrows indicate the actions taken by each agent's policy, coloured circles mark the final states (before the termination action is triggered), and the white circle denotes the estimated Fermat state corresponding to these final states. The bars on the right of each figure show the Fermat $n$-distance estimates for each feature. Please see Appendix A.7 for other state initializations.

is motivated by the collective nature of the options discovered, where effective option selection often depends on team-level information. In a scenario where agents must search for resources in an environment, a good strategy might be to spread out, locate a resource, and then trigger an option to gather around it. Without knowing if a teammate has found the resource, however, triggering the option prematurely can hinder performance and destabilise exploration.

**Assumption 2.** *There is a synchronisation mechanism that ensures the minimum number of agents required for the correct execution of joint options.*

In the same way that passing in a ball game cannot be defined without a receiving partner, this design choice synchronises joint option selection, asserting that enough agents agree on following an option at a given time for that option to be activated and executed correctly.

Inspired by *local options* (Amato et al., 2019), we define *joint options* via hierarchical agent histories. A joint option is a tuple $\mathcal{W} = \langle I_{\mathcal{W}}, \pi_{\mathcal{W}}, \beta_{\mathcal{W}} \rangle$, where $I_{\mathcal{W}} \subseteq H_M$ and $\beta_{\mathcal{W}} : H_M \rightarrow [0, 1]$ are defined over joint macro-level histories, and $\pi_{\mathcal{W}} = (\pi_{\mathcal{W}}^1, \ldots, \pi_{\mathcal{W}}^N)$ is a joint policy mapping joint primitive-level histories $H_A$ to actions. We define joint options as collective behaviours involving the entire team, and therefore require full team consensus for initiation. Joint options can be viewed as a direct extension of local options to multi-agent behaviours. Although global consensus restricts the expressivity of representable behaviours, we adopt it for simplicity. Therefore, despite our discovery method supporting behaviours over arbitrary agent subsets, we restrict ourselves to team-level options and leave the integration of options involving subsets of agents to future work.

To integrate joint options into the MacDec-POMDP framework, we redefine the macro-action set $M_i$ to include both joint and local options, treating primitive actions as local options with immediate termination. When agent $i$ selects an option $M_i$ at time step $t$, this selection is interpreted as a *vote* toward option initiation; a threshold of $N$ votes is required for joint options, and 1 vote for local options. If this value is reached, the option is executed and control is transferred to the option policy until termination, otherwise control is returned at $t + 1$. Next, we redefine macro-observations $\zeta_i$ to incorporate information shared by teammates. Dec-POMDP-Com (Oliehoek & Amato, 2016) extends the standard framework with an explicit communication protocol. Since communication is not our focus, we mimic this step by allowing agents to share their observations directly and leave a more general treatment of communication to future work.

## 4 EMPIRICAL EVALUATION

We structured our empirical evaluation around three hypotheses: **(H1)** Joint options provide advantages in downstream tasks compared with not using them. **(H2)** Joint options discovered via inter-agent relative representations (IARO) yield better downstream performance than those derived from other methods. **(H3)** The multi-dimensional $n$-distance representation enables a more robust option discovery process that its scalar-value variant in domains with more complex state spaces. Additionally, we investigate how increasing or decreasing the number of relative options discovered by our framework affects overall performance in our experimental domains.

**Experimental Setup.** We evaluated our approach in two multi-agent domains: Level-Based Foraging (Papoudakis et al., 2021) and Overcooked (Ruhdorfer et al., 2024), using their JAX re-

implementations from jum (2024) and Rutherford et al. (2024), respectively. We focused on two scenarios in each domain: for LBF, these are 15x15-4p-3f and 15x15-4p-5f and for Overcooked, they are Forced Coordination and Counter Circuit. In LBF, we introduced stronger coordination requirements by setting each apple's level equal to the sum of all agents' levels, the forced cooperation configuration from Papoudakis et al. (2021). Our choice of domains reflects a trade-off between interpretability and feature diversity. LBF enables straightforward visualisation of eigenvectors and relative representations through its $X, Y$-coordinate state space. Overcooked involves richer feature semantics, combining $X, Y$-coordinates, orientations, and a categorical variable for item inventory. While information sharing is implicit in the Overcooked task, in LBF, we allowed agents to always observe the relative distances to their teammates (rather than only when they enter their field of view) and add a flag to their observations indicating when each teammate is in the vicinity of an apple. Our approach and all baselines operated under the same level of observability in our analysis.

To train the $n$-distance encoder and the graph Laplacian eigenvector approximator ALLO (Gomez et al., 2024), we used a dataset of 500,000 transitions sampled from a random joint policy in each domain. In LBF, we approximated the first 10 eigenvectors, yielding 20 options, while in Overcooked we approximated 20 eigenvectors, yielding 40 options. Once estimated, we trained joint option policies based on these eigenvectors for one million steps, equivalent to 5% and 10% of the total training time in each task. We employed IQL to train the option policies and incorporated an action, $\mathcal{A}'_i = \mathcal{A}_i \cup \{\perp\}$, as the termination condition (Machado et al., 2017b). All agents involved had to all choose this action for an option to be terminated. In addition, we enforced a hard stop after 50 steps. The initiation set was defined as the entire joint state space, allowing options to be started anywhere, provided that the required number of agents, $n_{\mathcal{W}} = N$, was met.

**Evaluation against generic baselines.** To evaluate **H1**, we compared the performance of IQL equipped with *inter-agent relative* options (IQL+IARO) against four option-free baselines: MAPPO (Yu et al., 2022), IPPO, IQL, and VDN (Sunehag et al., 2018). Following Rutherford et al. (2024), we left MAPPO out of our analysis for Overcooked. Figure 4 presents IQM scores with 95% confidence intervals (CI) computed across 10 seeds. The addition of joint options (IQL+IARO) led to consistent performance gains over the vanilla IQL, achieving higher percentages of apples eaten per episode in LBF and more successful deliveries per episode in Overcooked; see top row of Figure 4. This improvement was especially visible in Overcooked, where IQL tends to remain stuck in suboptimal solutions. The joint options equipped agents with coordination skills that systematically explored the state space and enabled them to swiftly parse through various coordination patterns in search of better strategies. Additionally, IQL+IARO outperformed the other baselines across the set of experiments, except the Forced Coordination scenario where VDN is known to perform well (Rutherford et al., 2024). These results highlight the benefits of joint options in overcoming the limitations of independent policy learning in multi-agent tasks by encouraging cooperation through the set of pre-computed coordination behaviours; note that our method does not rely on centralization (MAPPO) or value decomposition (VDN) to support it. However, we did observe a slower convergence at the start of training, which we attribute to known challenges of training with options under *global* initiation sets (Jong et al., 2008; Machado et al., 2023).

**Evaluation against other option frameworks.** Next, to evaluate hypothesis **H2**, we compared the downstream benefits of the options discovered with our framework against those discovered through an existing Kronecker graph product method from Chen et al. (2022) (IQL+Kron), and an ablation where discovery was performed directly on raw joint states (IQL+RJS). For IQL+Kron, we closely followed the original implementation, apart from two main adaptations to match our framework: (i) we employed ALLO (Gomez et al., 2024) to approximate both eigenvectors and eigenvalues directly; and (ii) we used a single ROD cycle to estimate multiple eigenvectors. Figure 4 compares the resulting options across both domains and shows that the options discovered by our method better equip teams of agents with the cooperative skills necessary to solve these downstream tasks. We noticed that, particularly in LBF, options discovered via raw joint states and Kronecker products can degrade performance. We believe that this is due to their first non-trivial eigenvectors mainly capturing behaviours that drive agents to the edges of the state space (also see Appendix A.6), which is counterproductive for the apple-picking task.

We then explored **H3** by evaluating the utility of the multi-dimensional $n$-distance representation (IQL+IARO-MultiDim) against its scalar-value variant (IQL+IARO-Scalar). While in LBF, the domain with simpler state definitions, both methods performed similarly, in Overcooked, the multi-distance method proved more effective, as agent states consist of multiple features of diverse se-

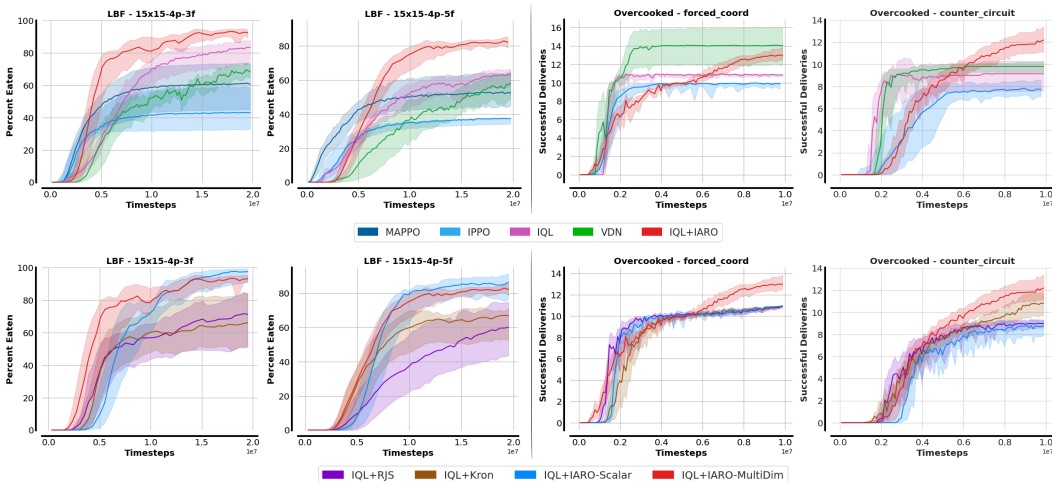

Figure 4: Downstream task performance analysis for both environments (LBF on the left, Overcooked on the right). The top row compares IQL+IARO against option-free baseline algorithms, while the bottom row compares it against IQL augmented with other option discovery methods.

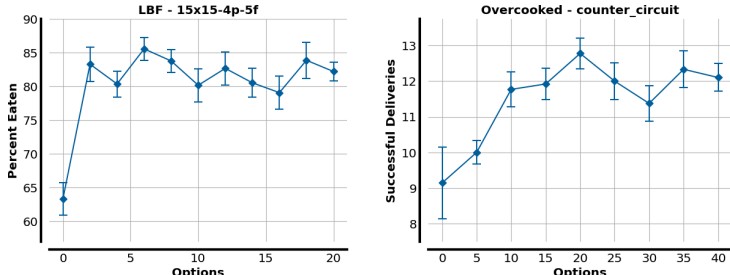

Figure 5: Downstream task performance for the most complex scenario in LBF and Overcooked, evaluated using different numbers of options. We report IQM scores over 15 seeds and 64 evaluation episodes at the end of training for each configuration, with standard deviations shown as error bars.

mantics, highlighting the benefits of the disentangled $n$-distance representation. By decomposing $n$-distance estimation across features, agents can align on specific subsets of state dimensions, yielding a richer set of cooperative behaviours. It is then up to the acting policy to decide, via exploration, which alignment strategies are beneficial for the task at hand.

A complementary episodic reward analysis for a larger set of domains is provided in Appendix A.4, while other implementation details and the list of hyperparameters can be found in Appendix A.10.

**Option count analysis.** We examined how the number of joint options used during training influences downstream task performance. Our goal was to identify the threshold at which a subset of options yields substantial performance gains, as well as to understand how further increasing the number of options affects results. Figure 5 reports the aggregated scores for the most complex scenario in both environments. We observed that both tasks benefit even from a relatively small option set. In LBF, the largest boost in apples collected appears with as few as two options, while notable improvements in successful deliveries emerge with the first ten options in Overcooked. This is consistent with eigenoption theory, which suggests that the first eigenvectors connect distant nodes in the state-transition graph, yielding powerful exploration behaviours. Later eigenvectors encode shorter time-scale behaviours, whose usefulness may vary in each task. We also note that more complex state spaces generally imply a larger threshold, reflecting the greater diversity of long-horizon alignment patterns that emerge through different feature combinations. Finally, although larger option sets can sometimes further improve performance (e.g., six options for LBF and twenty for Overcooked), these gains eventually saturate. Larger option sets can introduce increased training instability, resulting in reduced performance or higher variance. In our previous evaluations we utilised the maximum number of options for both environments to enable a fair comparison with the other option discovery methods.

## 5 RELATED WORK

We discuss related work on similar metrics, state representations, and temporally-extended actions.

**State similarity metrics.** Estimating state similarity measures is a fundamental challenge in RL. Even in simple domains, standard distances such as *Euclidean*, fail to capture true state proximity in the presence of obstacles, as they ignore environmental dynamics. In contrast, *bisimulation* metrics (Ferns et al., 2004) measure state similarity through differences in rewards and transition dynamics, and have been widely applied in optimality preserving state aggregation methods (Li et al., 2006). However, these approaches struggle in sparse-rewards settings, leaving them susceptible to representation collapse (Kemertas & Aumentado-Armstrong, 2021; Chen et al., 2024). *Temporal* or *successor* distances define state similarity as the expected number of actions required by a policy to travel between two states (Venkattaramanujam et al., 2019). This class of state distances is invariant to state representations and closely reflects environmental dynamics, making it a popular solution for goal-conditioned RL (Hartikainen et al., 2020; Durugkar et al., 2021; Myers et al., 2024), intrinsic reward composition (Bae et al., 2024; Jiang et al., 2025) and unsupervised skill discovery (Park et al., 2024). Notably, METRA (Park et al., 2024) is particularly relevant to our work, as it aims to learn skills that explore a latent space connected to the ground state space via a temporal metric. Besides performing skill discovery in single-agent settings, a main distinction is that we leverage temporal distances to approximate a latent representation for the eigenoption discovery of joint options that achieve exploration at different *time-scales*, rather than only at its extremes.

**State representations in MARL.** State representations in MARL have been mostly used for the aggregation of agents' local observations into a compact global representation, often through graph neural networks (GNNs). GNNs are invariant to the number of entities, here agent observation embeddings, and can weight the information passed between vertices differently through edge features (Jiang et al., 2020; Liu et al., 2019; 2021; Nayak et al., 2023). Utke et al. (2025) emphasise the importance of relative information for estimating inter-agent relations, and embed this inductive bias into GNN edge features based on hand-crafted spatial relations. In contrast, our method learns inter-agent relations automatically, without any restrictions on feature semantics.

**Temporally extended actions for multi-agent systems.** Makar et al. (2001) extend semi-Markov decision processes to cooperative multi-agent settings, introducing two execution schemes: synchronous (macro-actions terminate simultaneously) and asynchronous (macro-actions terminate independently). Asynchronous schemes gained recent popularity due to their innate generality (Amato et al., 2019; Xiao et al., 2021; 2022), but typically rely on single-agent (*local*) options that do not express cooperative behaviours. Therefore, coordination occurs only at the option selection level, but not within the option policies themselves. This approach drastically limits the expressiveness of the options used in multi-agent scenarios. To address this, Chen et al. (2022) integrates *option discovery* techniques based on *covering options* (Jinnai et al., 2019b) that construct joint behaviours via Kronecker products of single-agent transition graphs. However, the resulting joint options mainly synchronise independent behaviours, failing to capture strong inter-agent dependencies or correlations, a limitation noted by the authors. This leaves the problem of discovering strongly coordinated behaviours still open, which is precisely what we aim to address with our method.

## 6 CONCLUSION

In this work, we introduced a novel *inter-agent relative* representation for joint states, designed to address the key challenges of multi-agent option discovery. This representation compresses the joint state space and re-centres it around the point of maximum alignment for the team. We define this point as the *Fermat state* and propose a method that estimates it explicitly. Using the relative representation, we then produce joint options that are strongly coordinated and well-suited to capture inter-agent relational dynamics. Moreover, by disentangling the representation across individual state features, our approach further enriches the behavioural diversity expressed in the discovered joint options. We demonstrated the effectiveness of the proposed method across multiple benchmark domains and scenarios, confirming its ability to support agent teams in achieving stronger solutions on downstream tasks. Our work opens up multiple directions for further research in the discovery and use of options for multi-agent collaboration. In particular, we will study the discovery of richer topologies of coordinated behaviours, relax the assumption of sharing observations in lieu of a communication protocol, and the restriction that joint option initiation requires full team consensus.

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

# A    APPENDIX

## A.1    PRACTICAL EXAMPLES IN REALISTIC SCENARIOS

We present two example scenarios where identifying coordination strategies can drastically acceler-
ate the computation of effective solutions in downstream tasks: a search-and-rescue team operation
and the dynamics of a professional restaurant kitchen.

As the first example, consider a team of AI agents assisting humans in a search and rescue operation.
The appropriate strategy for the team depends on multiple factors, such as the type of terrain to be
explored (e.g., open fields versus dense forest), the capabilities of the agents involved (e.g., bipedal
robots or drones), and the level of effort required to assist each individual in need (e.g., assist in
extracting the human from rubble, or arranging for an ambulance). In dense forests, teams may need
to form tight lines or sweep formations to ensure perfect coverage, whereas open fields can often
be searched more efficiently by dispersion. Moreover, when victims require substantial assistance,
teams may need to alternate between broad search patterns and swift regrouping around targets to
deliver help quickly and effectively.

As another example of multi-agent collaboration, consider a professional kitchen with multiple AI
agents coordinating their activities, operating with maximum efficiency and precision to prepare and
serve high-quality dishes as quickly as possible. The complex dynamics of the scenario require both
division of labour and synchronisation, with different subsets of the agents pursuing coordinating
patterns for preparing ingredients and cooking the dishes while synchronising their activities at
specific steps in the recipe, and delivering the dishes to provide a good experience for the customers.

Humans often draw on their rich contextual understanding to identify and apply the correct strategy
for these (and other such) tasks, although even humans will undergo specialised training to acquire
this contextual understanding. On the other hand, learning complex, temporally-extended, and var-
ied coordination strategies from scratch is extremely challenging for teams of AI agents. Sparse
reward (i.e., feedback) signals, non-stationarities in the dynamics of the domain and the agents, and
credit assignment are among multiple reasons that make this learning particularly challenging. The
discovery of intermediate coordination patterns (*options*) that aid in the reliable and efficient com-
pletion of such tasks by AI agents is central to our efforts; our objective (in this paper) is to develop
a method capable of identifying these patterns from a limited number of interactions with the envi-
ronment. Such a method enables the team of agents to focus solely on learning when and where to
deploy each strategy, a substantially easier task in complex downstream scenarios. The domains we
use for experimental evaluation are simplified versions of the complex scenarios described above.

## A.2    MUTUAL INFORMATION OBJECTIVE PROOF

Next, we provide the proof of Proposition 1 introduced in Section 3.2.

**Proposition 1.** For any two agent indexes $i, j \in \mathcal{I}$, with $i \neq j$, and feature index $f \in \{1, \ldots, F\}$:

$$\mathbb{I}(S_{-f}^{i,j}; Z_f^{i,j}) \leq \mathbb{I}(S_{-f}^{i,j}; S_f^{i,j}) + \mathbb{I}(S_{-f}^{i,j}; Z_f^{i,j} \mid S_f^{i,j}).$$

*Proof.* Let $A$, $B$, and $C$ be three random variables such that $A, B, C \sim p(a, b, c)$, where $p(a, b, c)$ is
the joint distribution, and let $I(A; B, C)$ be the Multivariate Mutual Information (MI) estimate for
these variables. Then, from the chain rule of MI:

$$\mathbb{I}(A; B, C) = \mathbb{I}(A; C) + \mathbb{I}(A; B|C)$$
$$\mathbb{I}(A; B, C) = \mathbb{I}(A; B) + \mathbb{I}(A; C|B)$$

and therefore:

$$\mathbb{I}(A; C) + \mathbb{I}(A; B|C) = \mathbb{I}(A; B) + \mathbb{I}(A; C|B)$$

Given that an MI estimate is always positive, i.e. $\mathbb{I}(A, B|C) \geq 0$:

$$-\mathbb{I}(A; B|C) \leq 0$$
$$\mathbb{I}(A; C) - \mathbb{I}(A; B) - \mathbb{I}(A; C|B) \leq 0$$
$$\mathbb{I}(A; C) \leq \mathbb{I}(A; B) + \mathbb{I}(A; C|B)$$

Replacing $A = S_{-f}^{i,j}, B = S_f^{i,j}$ and $C = Z_f^{i,j}$, we obtain the inequality in Proposition 1 (Equation 5).

## A.3 CMI MINIMISATION ALGORITHM

Algorithm 1 follows the framework of Dunion et al. (2023) to minimise the CMI estimator defined in Equation 6. The algorithm first samples a batch of state pairs from the same time step in a trajectory and encodes their multi-dimensional distance. For each pair, it iterates over state features and forms a conditioning term by concatenating the feature value with its corresponding distance component. It then selects the most similar neighbours in the batch (based on this concatenated representation) and constructs negative samples by shuffling the remaining feature values, while the original state pair serves as the positive example. The $n$-distance estimator is trained adversarially to minimize a discriminator's ability to distinguish positive from negative pairs, thereby reducing correlations between feature values belonging to the same state.

---

**Algorithm 1** CMI minimisation step

---

**Require:** Transitions $\tau = \{s_0, \ldots, s_T\} \sim \rho_\pi$, joint-state factorisation function $g : \mathcal{S} \to \prod_{i=1}^N \mathcal{S}^*$.
**Require:** Parameters for the distance estimator $\theta$ and the discriminator $\psi$.
1: Factorise each joint state into N single-agent states $\{s_t^0, \ldots, s_t^N\} = g(s_t)$, with $t \in \{0, \ldots, T\}$.
2: Create single agent state pairs $b_t^{i,j} = \{(s_t^0, s_t^1), (s_t^0, s_t^2), \ldots (s_t^{N-1}, s_t^N)\}$.
3: Concatenate the single agent state pairs into the final batch for CMI minimisation:
$$B^{i,j} = \{(s_0^0, s_0^1), \ldots (s_0^{N-1}, s_0^N), (s_1^0, s_1^1), \ldots (s_T^{N-1}, s_T^N)\}$$
4: Initialise $\mathcal{L}_D \leftarrow 0$ and $\mathcal{L}_A \leftarrow 0$.
5: Forward pass through multi-feature distance encoder $z_n = d_\theta^F(s_n^{i,j})$, where $s_n^{i,j}$ is the $n$-th pair in the single agent dataset $B^{i,j}$.
6: **for** $n \in (1, \ldots, |B^{i,j}|)$ **do**
7:     **for** $f \in (1, \ldots, F)$ **do**
8:         Create conditioning set $c_{f,n} = (s_{f,n}^{i,j}, z_{f,n})$.
9:         Find $k$ nearest neighbours (kNN) of $c_{f,n}$ in the batch: $\sqrt{\sum_i \left((c_{f,n})^i - (c_{f,n'})^i\right)^2}$.
10:         Create $s_{-f,n}^{\text{perm}} = \{s_0^{\text{perm}}, \ldots, s_{f-1}^{\text{perm}}, s_{f+1}^{\text{perm}}, \ldots, s_F^{\text{perm}}\}$ by shuffling the kNNs.
11:         Calculate discriminator loss:
$$\mathcal{L}_D \leftarrow \mathcal{L}_D + \log \sigma(D_\phi(s_n^{i,j}, z_{f,n}) + \log\left(1 - \sigma(D_\phi(s_{-f,n}^{\text{perm}}, s_{f,n}^{i,j}, z_{f,n}))\right)$$
12:     **end for**
13: **end for**
14: Update discriminator parameters to minimise $\mathcal{L}_D$.
15: **for** $n \in (1, \ldots, N)$ **do**
16:     **for** $f \in (1, \ldots, F)$ **do**
17:         Calculate adversarial loss: $\mathcal{L}_A \leftarrow \mathcal{L}_A + \log\left(1 - \sigma(D_\phi(s_n^{i,j}, z_{f,n}))\right)$.
18:     **end for**
19: **end for**
20: Update encoder parameters to minimise $\mathcal{L}_A$.

---

## A.4 EPISODIC REWARD ANALYSIS & ADDITIONAL RESULTS

Figure 6 presents an extended analysis of our method using episodic returns as the evaluation measure. Beyond the four scenarios discussed in Section 4, we add one additional setting for each domain: 15x15-3p-5f in Level-Based Foraging (LBF) and *Asymmetric Advantages* in Overcooked. These scenarios, together with the original four, were selected because the IQL baseline finds it difficult to compute solutions for these scenarios (Papoudakis et al., 2021; Rutherford et al., 2024), thereby highlighting the improvements achieved by our approach. Similar to Figure 4, we report the IQM scores with 95% CI for 10 seeds.

## A.5 $n$-DISTANCE REPRESENTATION COMPARISON

Figure 7 compares the outputs of the trained $n$-distance estimator, conditioning on the positions of two agents, for the scalar and multi-dimensional variants. Since the two state features, $X$ and

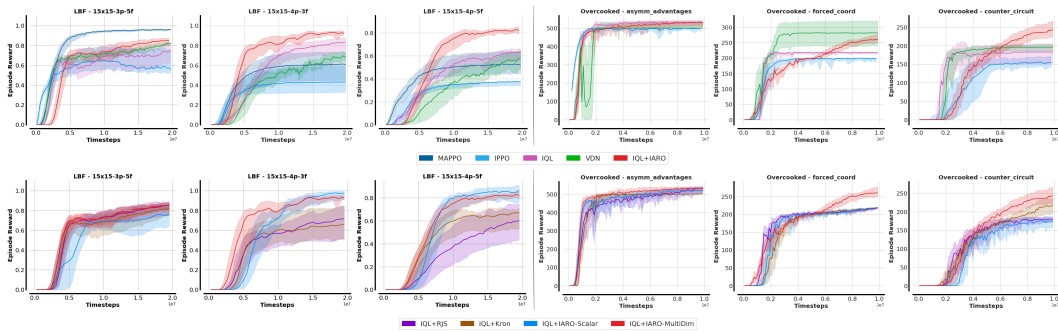

Figure 6: Episodic reward IQM results for the entire suite of environments (including 15x15-3p-5f for LBF and Asymmetric Advantages for Overcooked).

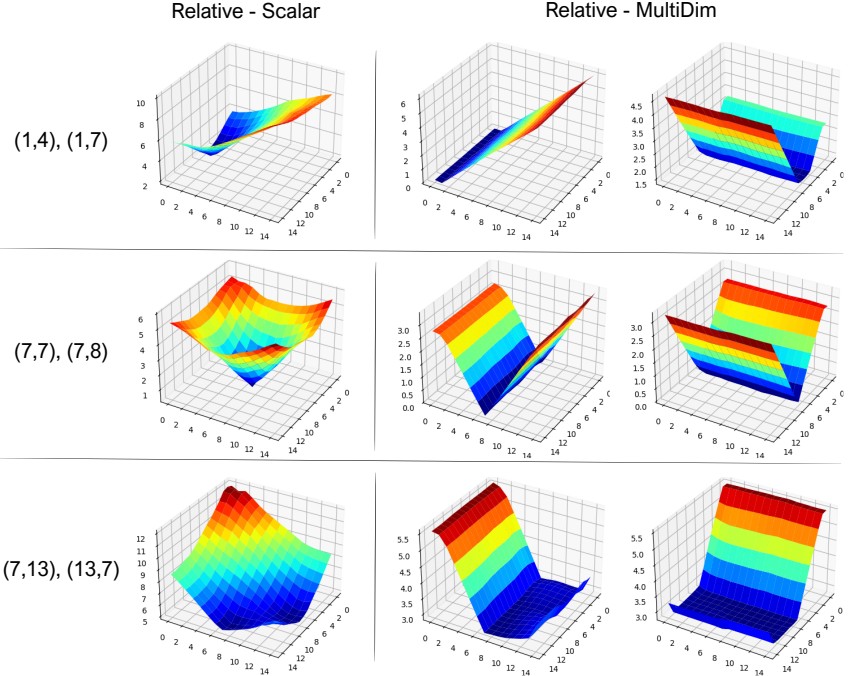

Figure 7: A visualisation of the $n$-distance approximator outputs for the scalar and multi-dimensional disentangled variants in a $15{\times}15$ grid environment with three agents, where we fix the positions of two agents and show the $n$-distance values as the position of the third agent varies.

$Y$ coordinates, are independent and no obstacles are present, the multi-agent temporal distance approximates a measure similar to a standard spatial distance. The scalar estimator computes the $n$-distance jointly across both axes, whereas the multi-dimensional variant disentangles them by applying the proposed MI penalty. At the centre of these representations lies the *Fermat* state, as a point of minimal distance from the fixed coordinates of the teammates.

## A.6 EIGENVECTOR COMPARISON

In this section, we offer a more detailed explanation of the differences between eigenvectors computed on the inter-agent relative representation of the joint state space, and those obtained by applying eigenoption discovery directly on the raw states. We also extend the visualisation in Figure 2 to incorporate the Kronecker eigenvector (Chen et al., 2022) approximations, and the eigenvectors obtained from both the scalar and multi-dimensional $n$-distance representations presented in Figure 7, not just the latter.

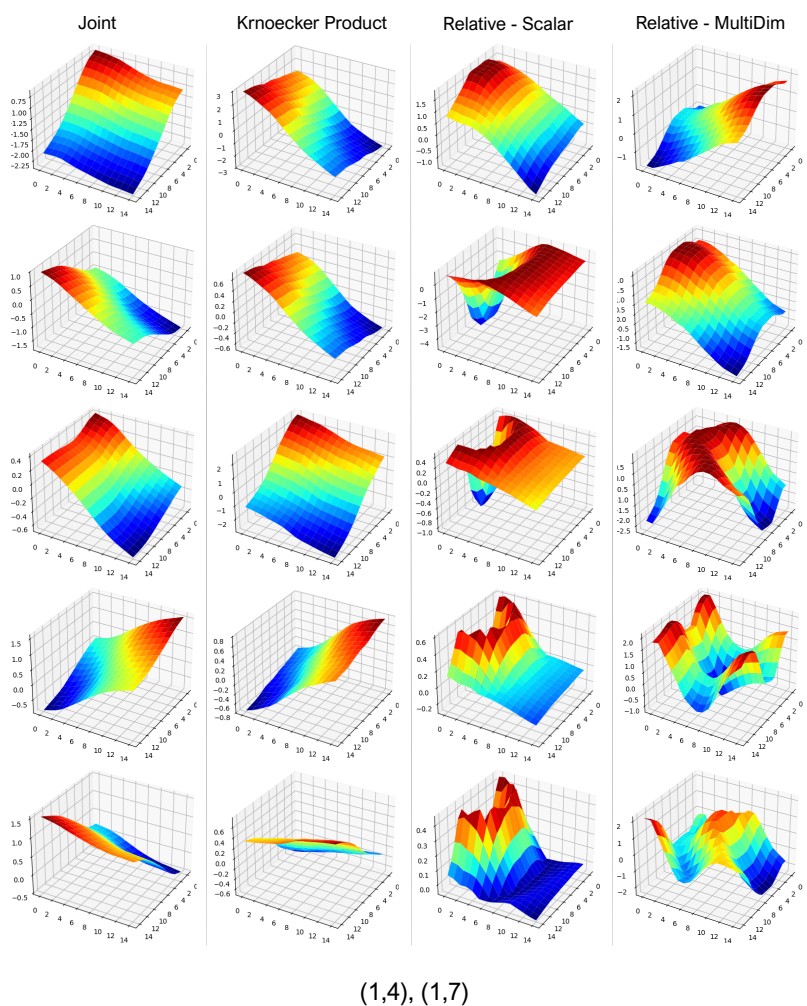

(1,4), (1,7)

Figure 8: The first five non-trivial eigenvectors resulted from eigenoption discovery on raw joint states, through the Kronecker product of single agent eigenvectors, and the two proposed inter-agent relative representation-based methods. The conditioning states for the first two agents in the team are the coordinates (1,4) and (1,7).

To this end, we fix two agents at specific positions and examine the values of each eigenvector from the perspective of the remaining agent. Under the assumption of agent homogeneity, agents are interchangeable, and examining the perspective of one agent yields meaningful insight into the team-level subgoals captured by the eigenvectors. We discuss the heterogenous case in detail in Section A.8. This statement holds in particular for eigenvectors that encode coordinated behaviours, whereas the choice of conditioning order can lead to different results for more independent behaviours. We found this to be evident in the Kronecker product approximation, where joint eigenvectors are formed as products of multiple single-agent eigenvectors.

Figures 8-10 provide a more extensive analysis of the first five non-trivial eigenvectors for each of the four frameworks: Eigenoption discovery on raw joint states (Joint), Kronecker product-based joint option discovery (Kronecker Product), and the two proposed inter-agent relative options (Relative-Scalar and Relative-MultiDim). We also extend the analysis to three distinct conditioning pairs for the positions of the first two agents: [(1,4),(1,7)], [(7,7),(7,8)], and [(7,13),(13,7)]. Please note the negated version of each eigenvector, as the eigenvectors produce two options with opposite effects.

The key difference between eigenvectors derived from the non-relative and relative representations is the lack of responsiveness of the Joint and Kronecker product eigenvectors to different conditioning pairs. This effect is pronounced for the Kronecker product eigenvectors, which remain

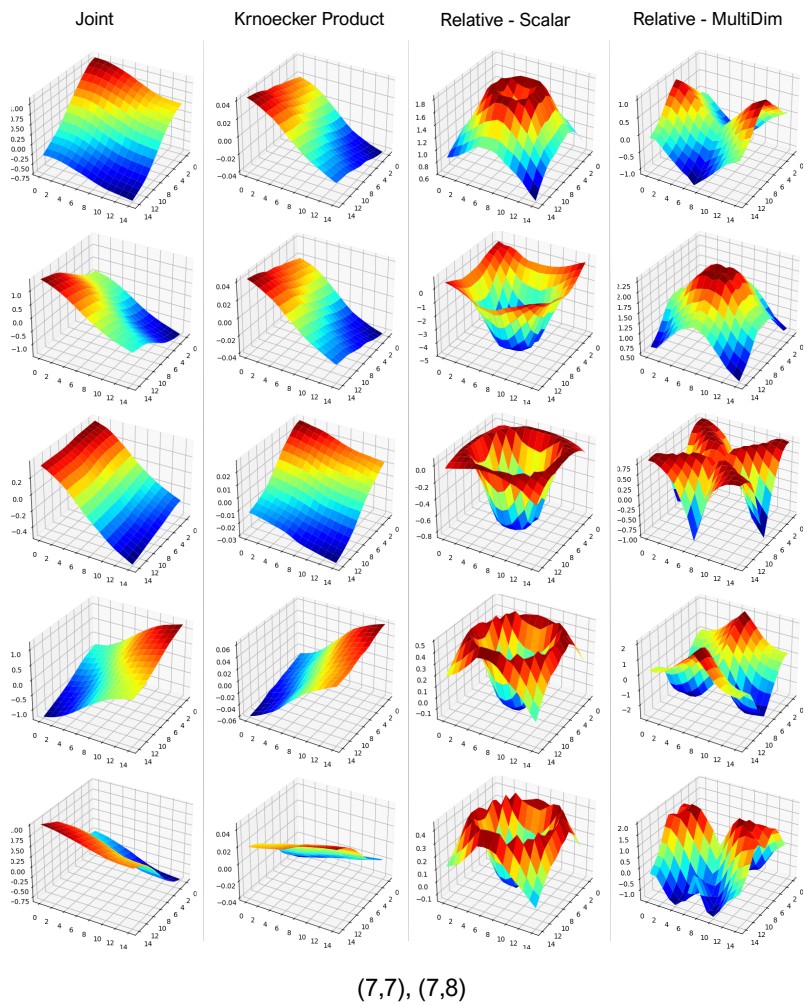

(7,7), (7,8)

Figure 9: The first five non-trivial eigenvectors resulted from eigenoption discovery on raw joint states, through the Kronecker product of single agent eigenvectors, and the two proposed inter-agent relative representation-based methods. The conditioning states for the first two agents in the team are the coordinates (7,7) and (7,8).

completely unaffected by teammate positions, further validating the absence of interdependence or coupling in the behaviours captured by this method. This effect arises because both the raw joint state and Kronecker product methods emphasise exploration of the state space rather than inter-agent relations. In contrast, by centring the representation around the *Fermat* state, the relative representation yields behaviours that adapt to diverse team configurations and produce various patterns of agent alignment. While the scalar representation yields behaviours that align the agents at different distances to each other (on both axes), the multi-dimensional representation enables various in-phase and off-phase behaviours to be discovered with different combinations of features. In addition, as we proceed through the set of estimated graph Laplacian eigenvectors, individual elements capture progressively shorter time scales. This is a consequence of the orthogonality constraint imposed by estimating multiple eigenvectors per cycle while using graph-drawing–based objectives, ALLO (Gomez et al., 2024).

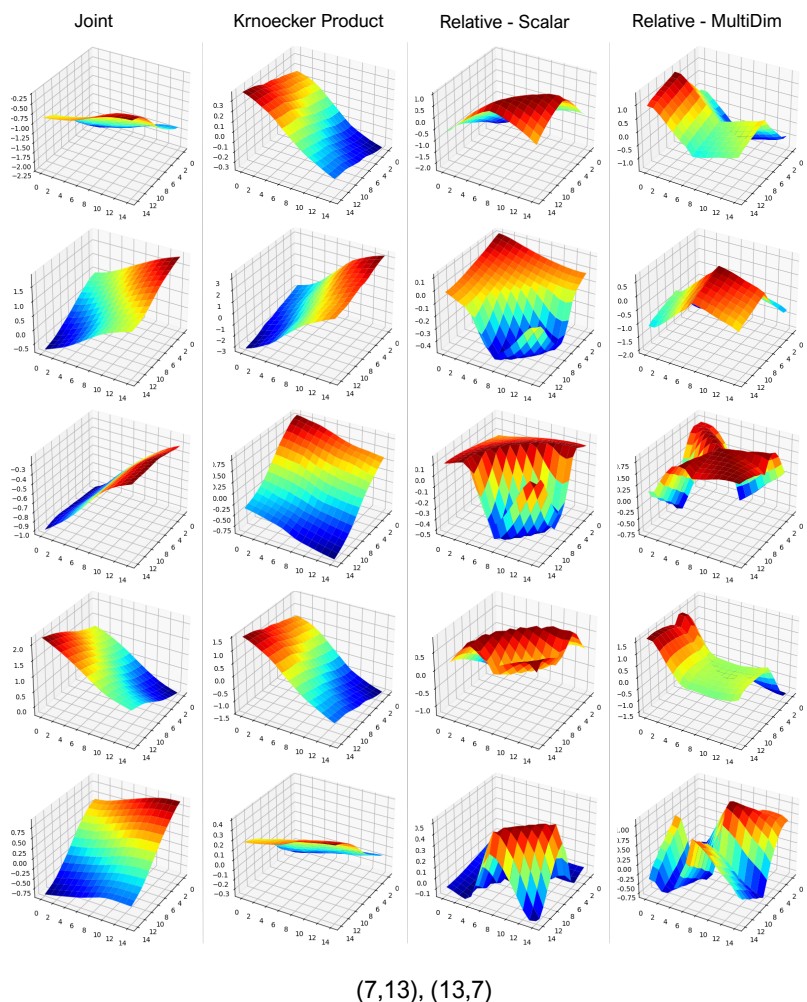

(7,13), (13,7)

Figure 10: The first five non-trivial eigenvectors resulted from eigenoption discovery on raw joint states, through the Kronecker product of single agent eigenvectors, and the two proposed inter-agent relative representation-based methods. The conditioning states for the first two agents in the team are the coordinates (7,13) and (13,7).

## A.7 OPTION POLICY VISUALISATION

Figure 11 presents rollouts for the first four options, corresponding to the first two relative multi-dimensional eigenvectors described in Section A.6. Each row depicts a different agent initialization, illustrating how the behaviours dynamically adapt to changes in the coordinates of the Fermat state. The bars on the right of each rollout indicate the $n$-distance of the final joint state, highlighting the ability of the learned options to selectively align agents along specific state features.

## A.8 HETEROGENEOUS STATE SPACES

In this section, we present our extension of the homogeneous state space framework (Section 3) to heterogeneous state spaces. Due to the need for meaningful state-space alignment, we focus on scenarios in which each agent shares a subset of its state features with one or more teammates. Without such shared features, state synchronisation is not a valid strategy for coordination. Furthermore, we expect each agent to be able to infer its teammates' types from its observations, either explicitly, when the observations contain type information, or implicitly, for example by enforcing a clear ordering of features or by using fixed positions in the observation vector. We believe these expecta-

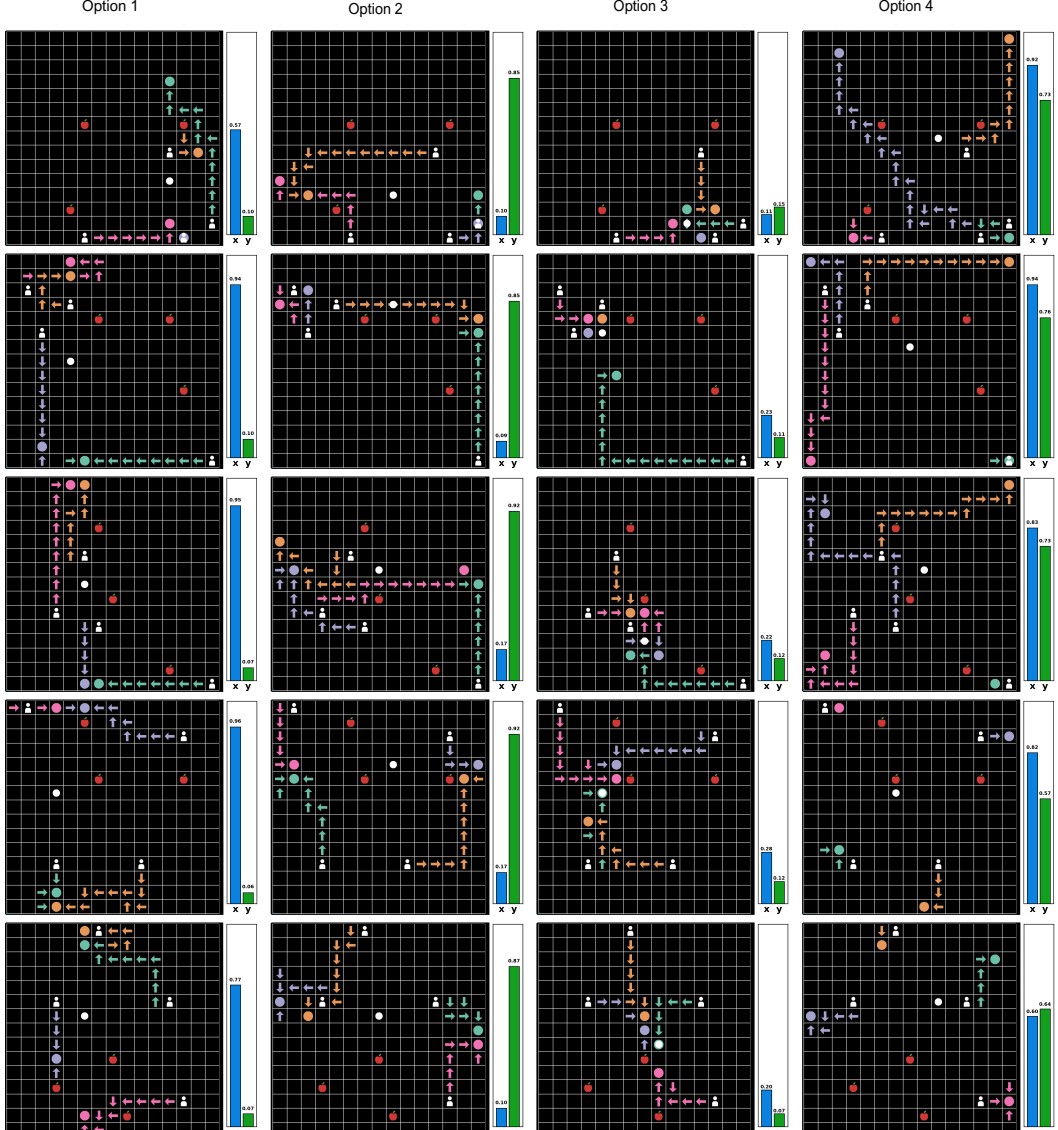

Figure 11: Visualisation of the policy roll-outs for the first four learned options in the 15×15 grid environment with four agents, illustrating four distinct state-alignment patterns for multiple initial states. Arrows indicate the actions taken by each agent's policy, colored circles mark the final states (before termination action is triggered), and the white circle denotes the estimated Fermat state corresponding to these final states. The bars on the right of each figure show the Fermat $n$-distance estimates for each feature.

tions are representative of practical multi-agent collaboration domains, and point to agent modelling techniques (Rabinowitz et al., 2018) as a potential way to relax the latter requirement.

Note that only two modules in our approach operate directly on state information, the pairwise state-distance function $d^F : \mathcal{S}^* \times \mathcal{S}^* \to \mathbb{R}^F$ and the Fermat encoder $\phi : \mathcal{S} \to \mathcal{S}^*$. In the homogeneous setting, we had $\mathcal{S}^* = \mathcal{S}^1 = \ldots = \mathcal{S}^N$, but this assumption does not hold in heterogeneous domains, thereby invalidating the original formulations of these two modules. To address this issue, we introduce the unified feature space $\mathcal{F}^* = \bigcup_{i=1}^{N} \mathcal{F}^i$, where $\mathcal{F}^i$ denotes the feature set underlying agent $i$'s individual state space $\mathcal{S}^i$, indicating which components of the factorised joint state space it includes. For each feature $f \in \mathcal{F}^*$, let $\mathbf{F}_f$ denote its domain, defined as the union of all values that feature can take across the state spaces of all agents. We then redefine the shared state space as the Cartesian product over all feature domains in the unified feature space, i.e., $\mathcal{S}^* = \times_{f \in \mathcal{F}^*} \mathbf{F}_f$. This definition

of $\mathcal{S}^*$ can represent each individual state space $\mathcal{S}^i$ by padding the features that appear in $\mathcal{F}^*$ but not in $\mathcal{F}^i$ with default values, e.g. zeros. Given the unified single agent state-space $\mathcal{S}^*$, we can retain our original definition of $d^F$ and $\phi$; as long as $d^F$ can identify the common features between two heterogeneous agent states and compute their similarity while ignoring any padded values, the rest of our proposed framework can remain unchanged. While a full treatment of heterogeneous state distance metrics is beyond the scope of this work, we outline (below) a set of guidelines for adapting the temporal state distance approach used throughout this paper to heterogeneous state spaces.

In Section 3, we rely on the temporal successor distance method of Myers et al. (2024) for computing pairwise state distances. We now outline how this method can be extended to heterogeneous state spaces through two main changes:

1. The MRN distance module is provided the source agent's type as an additional input.

2. We modify the sampling of goal states to include randomly shuffled values from states of other agent types, therefore achieving conditional independence to these features, given the source agent's type.

In the original MRN architecture (Liu et al., 2023), the distance encoder is computed as a sum of symmetric and asymmetric parts: $d_\theta(x, y) = \Delta(h_\theta(x) - h_\theta(y)) + \|g_\theta(x) - g_\theta(y)\|$, where $\Delta(x) = \max_{i=1}^d [\max(0, x_i)]$ and $h_\theta$ and $g_\theta$ represent the two halves of the outputs of a encoder network parameterised by $\theta$. In our adaptation, we extend this expression to include the agent type as follows. Let $s_t^i$ and $g_t^i$ be the source and goal state for agent $i$ at time step $t$, and $l^i$ the indexed type of this agent. Then, we modify the above expression for heterogeneous states as: $d_\theta(s_t^i, g_t^i, l^i) = \Delta\left(h_\theta(s_t^i, l^i) - h_\theta(g_t^i, l^i)\right) + \left\|g_\theta(s_t^i, l^i) - g_\theta(g_t^i, l^i)\right\|$. When $s_t^i$ and $g_t^i$ correspond to the same state type, $l^i$ is redundant. However, as stated in our second proposed change, the padded features of $g_t^i$, $\{f | f \in \mathcal{F}^* \wedge f \notin \mathcal{F}^i\}$, are randomly sampled from the goal states of agents of different types to $i$, resulting in the conditional invariance discussed above.

**Empirical analysis.**  To demonstrate the effectiveness of this extension, we constructed two heterogeneous toy scenarios by adapting the LBF environment to include two agent types: Type 1 (X axis only) agents and Type 2 (X & Y axes) agents. Type 1 agents' states only contain the X-axis coordinate, and their action space no longer includes the "Left" and "Right" horizontal movement actions. Type 2 agents have full $(x, y)$ state representations and can move along both axes, akin to the agents in the original environment.

Our experiments use two domain configurations. The first is a 10x10 grid containing one Type 1 and one Type 2 agent. The second is a 15x15 grid with three agents, where one agent is Type 1 and the remaining two are Type 2. The first configuration aims to analyse whether agents of different types can successfully identify and synchronise solely on their common features, while the second configuration examines whether selective alignment can be achieved between multiple agents that share different numbers of features.

Figure 12 illustrates the eigenvectors learned using our proposed heterogeneous extension for the two distinct agent types in the first configuration (10x10 grid). Since the two agents share only a single feature, the eigenvectors are identical from both perspectives, resulting in behaviours that align the agents solely along that feature. The lower part of the figure shows roll-outs from the first three option policies derived from the eigenvectors. The first option perfectly aligns the agents along the shared X-axis coordinate while completely disregarding the Y-axis. The second option aligns them along the Y-axis, ignoring the X-axis. The third policy positions the agents at a specific distance along the same axis, demonstrating behaviours at different time scales.

Figure 13 presents a similar analysis for the second configuration (15x15 grid). With the addition of another agent of Type 2, the eigenvectors enable these two agents to align along both axes of movement, while considering the third agent only with respect to the shared feature. This demonstrates the ability of the state alignment to occur selectively based on agent types. Specifically, the second eigenvector aligns an agent precisely with the Y-axis coordinate of its same-type teammate, while the third eigenvector captures behaviours that align all agents along the X-axis but only the corresponding ones (Type 2) along the Y-axis. We further support this observation by showing roll-outs from the trained option policies, which clearly exhibit these behaviours.

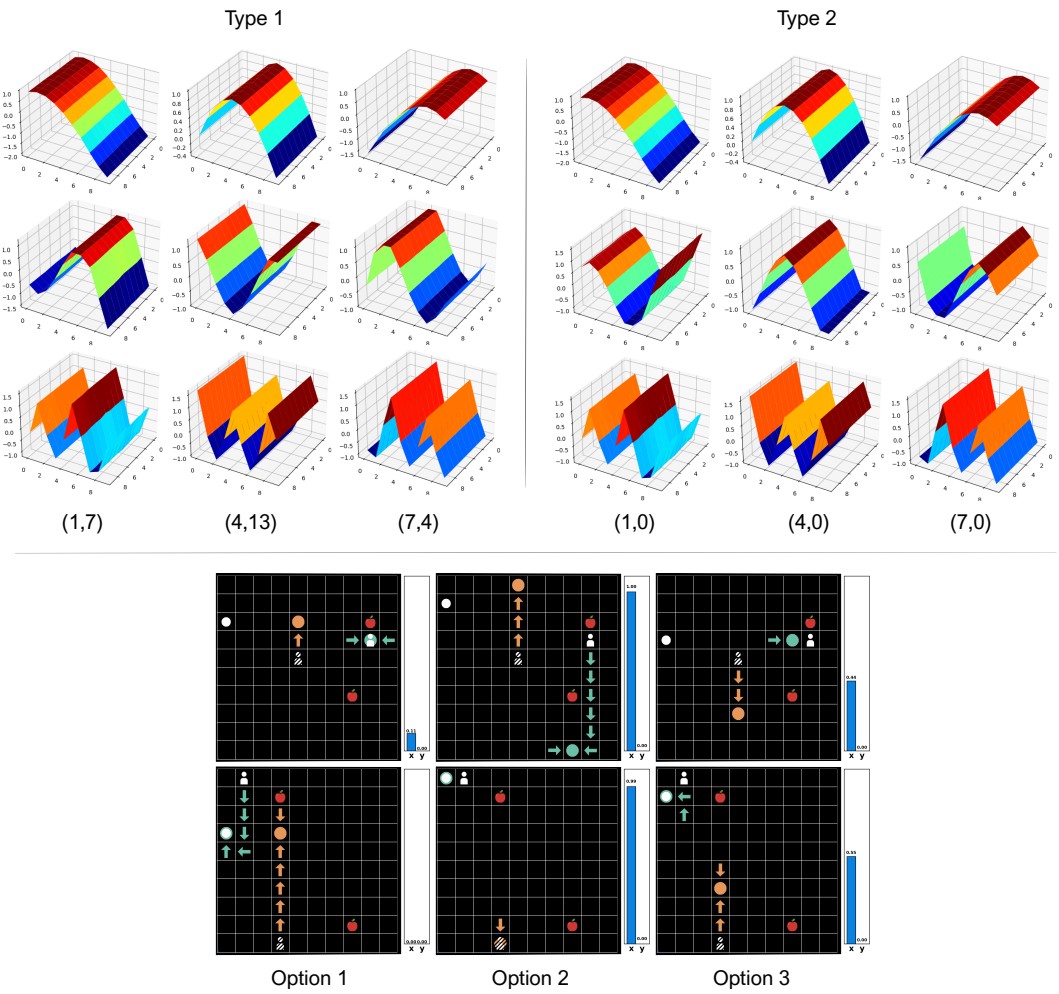

Figure 12: The first three eigenvectors, along with the first three options obtained using the proposed heterogeneous approach for a 10x10 grid with two agents of distinctive types: Type 1 and Type 2. The top part of the figure illustrates the eigenvectors, conditioned on the teammate's position at different coordinates, while the bottom part shows the roll-outs for the first three options generated. With padding, the Y-coordinate of the Type 1 agent is replaced with the value 0. We use diagonal stripes to identify the Type-1 agent in the grid.

## A.9 ROD CYCLE DISCUSSION

We adopt the original ROD cycle of Machado et al. (2017a) to approximate eigenoptions, estimating multiple eigenvectors within a single cycle. This design ensures orthogonality among eigenvectors, a property that enables options to function at different time scales. By contrast, other frameworks extract only one eigenvector per cycle, requiring multiple cycles to generate a full set of options, as in covering options (Jinnai et al., 2019b; Chen et al., 2022) and covering eigenoptions (CEO) (Machado et al., 2023). Covering options provide exploration guarantees by leveraging the Fiedler eigenvector to produce policies that connect the farthest points in the state-transition graph, an objective where orthogonality plays little role (Jinnai et al., 2019b). Our focus, however, is on discovering sets of cooperative behaviours that express diverse alignment patterns. In our experiments with CEO, the most recent framework, we found the resulting eigenvectors to exhibit limited diversity, see Figure 14, reinforcing our decision to return to the original approach.

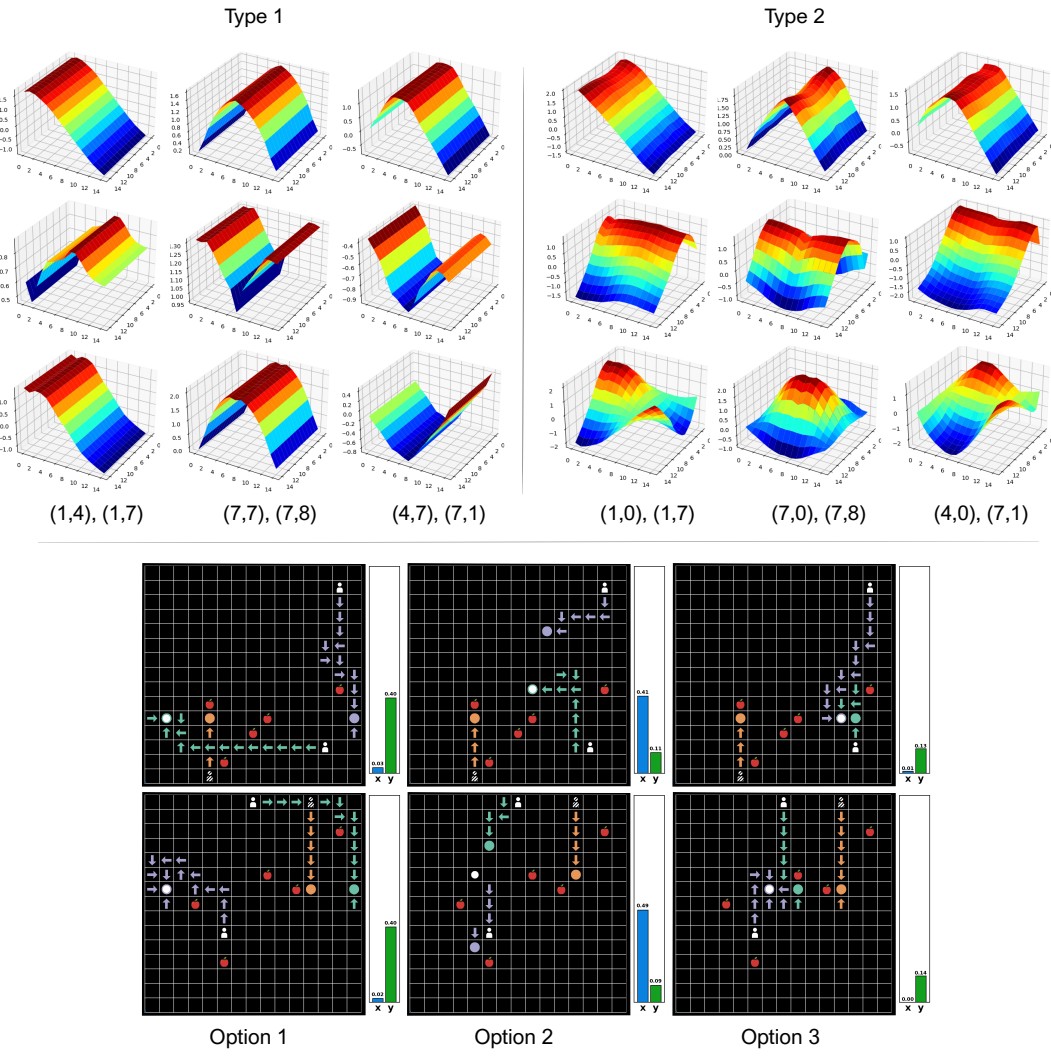

Figure 13: The first three eigenvectors, along with the first three options obtained using the proposed heterogeneous approach for a 15x15 grid with three agents of distinctive types: one agent of Type 1 and two agents Type 2. The top part of the figure illustrates the eigenvectors, conditioned on the teammate's position at different coordinates, while the bottom part shows the roll-outs for the first three options generated. With padding, the Y-coordinate of the Type 1 agent is replaced with the value 0.We use diagonal stripes to identify the Type-1 agent in the grid.

## A.10 IMPLEMENTATION DETAILS & HYPERPARAMETERS

Our implementations for the proposed discovery methods, baselines, and visualisations can be found at: https://github.com/raulsteleac/IARO.

$n$-**distance training.** We followed the publicly available implementation of *successor distances* from Myers et al. (2024) when integrating the CMD-1 architecture into our codebase. The temporal distance encoder $d_\theta$ is trained using the symmetrised InfoNCE contrastive loss (without resubstitution) (van den Oord et al., 2019), as suggested in the original work. We jointly train this encoder with the Fermat encoder $\phi$ using joint states sampled from a random joint policy and factorised as described in Section 3. Positive pairs for the InfoNCE loss are constructed by pairing current and future states from each agent's trajectory individually, without cross-agent mixing, while negatives are generated through in-batch shuffling, allowing combinations of states from different agents. To train the Fermat encoder, we incorporate $d_\theta$ into the objective in Equation 4 using a stop-gradient operator. When using a multi-dimensional $n$-distance $d_\theta^F$, we insert a lightweight linear projection to

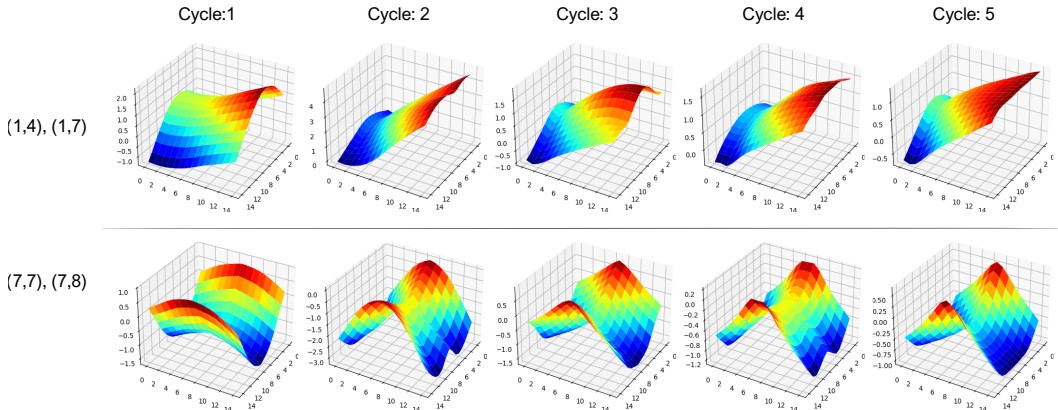

Figure 14: The eigenvectors generated by five consecutive CEO cycles, where once discovered, the option is introduced in the action space of the agents for exploration in the next phase.

map outputs to a scalar for contrastive loss computation. This projection is discarded after training, restoring the full multi-dimensional relative states. For Fermat encoder training, we instead sum the distance dimensions, since the errors on each dimension are weighted equally for this step.

In some scenarios, e.g., Overcooked Forced-coord and Asymmetric Advantages, agents cannot access the same states, as they are separated by walls. This causes the temporal distance estimator to encounter single-agent state combinations not present in training, resulting in noisy predictions. To mitigate this, we omit the feature causing the heterogeneity (the $Y$ coordinate) during state factorization. We note that this issue is tied to the temporal distance model itself, and our framework would operate successfully with any distance function not affected by this problem. We present the full list of hyperparameters for $n$-distance estimation training in Table 1.

| Hyperparameter | LBF | Overcooked |
|---|---|---|
| Distance encoder learning rate | 0.001 | 0.00005 |
| Fermat encoder learning rate | 0.001 | 0.00005 |
| Optimizer | Adam (Kingma & Ba, 2017) r | Adam (Kingma & Ba, 2017) |
| Distance encoder hidden layers | [256, 256] | [256, 256] |
| Distance encoder dimension (per feature) | 8 | 12 |
| Fermat encoder hidden layers | [256, 256] | [256, 256] |
| Minibatch size | 100 | 100 |
| # of epochs | 10 | 10 |
| Discriminator (CMI) learning rate | 0.0003 | 0.0001 |
| Discriminator hidden layers | [256, 256] | [256, 256] |
| # kNNs | 15 | 15 |
| Penalty weight | 0.003 | 0.0003 |

Table 1: Hyperparameters for training the $n$-distance estimator, for the scalar variant (up to the horizontal line) and the multi-dimensional variant (the entire set).

**ALLO training (Eigenvector approximator).** We used the same dataset to train the eigenvector encoder ALLO as for $n$-distance training. We followed the publicly available implementation referenced in Gomez et al. (2024), with similar hyperparameter configurations: two layers of 256 dimensions and barrier coefficient initiation value 2.

**Joint option policy training.** For joint option policy training, we used separate IQL architectures for each eigenvector sign (positive and negative), without parameter sharing, to accommodate potentially distinct agent behaviours. The environmental reward is thus replaced with $r_e(s, s') = e[s'] - e[s]$, for each eigenvector $e$ and two subsequent states $s, s'$. Table 2 lists the hyperparameters used for option policy training.

| Hyperparameter | LBF | Overcooked |
|---|---|---|
| Learning rate | 0.001 | 0.001 |
| Anneal learning rate | False | False |
| Optimizer | Adam (Kingma & Ba, 2017) r | Adam (Kingma & Ba, 2017) |
| Hidden layers | [64, 64] | [32,32] |
| CNN features | - | [16,16,16] |
| CNN Kernel dims | - | [[5,5], [3,3], [3,3]] |
| Parallel environments | 32 | 16 |
| Rollout steps | 10 | 10 |
| $\gamma$ | 0.99 | 0.99 |
| Buffer size | 5000 | $10^5$ |
| Buffer batch size | 32 | 128 |
| Target update interval | 10 | 10 |
| Maximum gradient norm | 1 | 10 |
| $\epsilon$ start | 1.0 | 1.0 |
| $\epsilon$ decay | 0.1 | 0.1 |
| $\epsilon$ finish | 0.05 | 0.05 |
| $\epsilon$ evaluation | 0.05 | 0.05 |
| Learning starts at timestep | 5000 | 1000 |
| # of epochs | 4 | 4 |
| # training steps | $10^6$ | $10^6$ |

Table 2: Hyperparameters used for training the joint opinion policies.

**MacDec-POMDP training.** We integrated the discovered set of options as additional actions in the original action space of each individual agent. For the backbone IQL implementation, we used an RNN decentralised architecture, trained with the set of hyperparameters presented in Table 3. We used the same list of hyperparameters when training IQL enhanced with options generated from each option discovery framework. To support the training of the decentralised policy over options, we additionally integrated previous SMDP training techniques like training primitive actions on option policy steps, intra-option learning and option interruption (Sutton et al., 1999). When integrating intra-option learning, we only reused the experiences of the executing joint option to train others when there was an exact match for the actions of each individual agent. For option interruption, we follow the *termination-improvement theorem* from Sutton & Barto (1998) and enable an option $w$ to be interrupted if $Q_i(H_M^i, \mathcal{W}) < V(H_M^i)$, for any agent $i$ currently following that option.

## A.11 LLM USAGE DECLARATION

In this work, the use of LLMs was kept to a minimum, providing limited support in writing tasks such as grammar correction and synonym search.

| Hyperparameter | LBF | Overcooked |
|---|---|---|
| Learning rate | 0.0005 | 0.0005 |
| Anneal learning rate | True | True |
| Optimizer | Adam (Kingma & Ba, 2017) r | Adam (Kingma & Ba, 2017) |
| Hidden layers | [128, 128] | [128, 128] |
| CNN features | - | [32,32,32] |
| CNN Kernel dims | - | [[5,5], [3,3], [3,3]] |
| Parallel environments | 32 | 16 |
| Rollout steps | 20 | 10 |
| $\gamma$ | 0.99 | 0.99 |
| Buffer size | 5000 | $10^5$ |
| Buffer batch size | 128 | 128 |
| Target update interval | 10 | 100 |
| Maximum gradient norm | 1 | 10 |
| $\epsilon$ start | 1.0 | 1.0 |
| $\epsilon$ decay | 0.1 | 0.1 |
| $\epsilon$ finish | 0.05 | 0.05 |
| $\epsilon$ evaluation | 0.05 | 0 |
| Learning starts at timestep | 5000 | 1000 |
| # of epochs | 4 | 4 |
| # training steps | $2 \times 10^7$ | $10^7$ |
| # step limit for option execution | 50 | 50 |

Table 3: Hyperparameters used for Mac-DecPOMDP training.

