# OpenReview forum: "Inter-Agent Relative Representations for Multi-Agent Option Discovery"
_ICLR.cc/2026/Conference — ICLR 2026 Poster_

### Official Review · Reviewer_KghA · 2025-10-26

**Soundness:** 3
**Presentation:** 3
**Contribution:** 3
**Rating:** 6
**Confidence:** 3

**Summary:**

This paper targets at multi-agent option discovery. Specifically, they propose a joint-state abstraction that compresses the state space while preserving the information necessary to discover strongly coordinated behaviors.

**Strengths:**

- Multi-agent option is an important and challenging research problem.
- This paper is tecnically solid and theretically grouned.
- The comparison with related work is comprehensive.

**Weaknesses:**

- MAPPO results are not provided for overcooked tasks.
- Please provide a discussion on the computation cost compared to other methods.
- Please provide a discussion on the compatibility with mainstream MARL methods, such as. VDN and MAPPO. Can they be integrated together?

**Questions:**

See weaknesses.

---

> ### Author Response · Authors · 2025-11-25
>
> We thank the reviewer for their comments and are grateful that they found our work “technically solid and theoretically grounded”, while also recognizing the importance of the research problem we aimed to address.
>
> > W1. MAPPO results are not provided for overcooked tasks.
>
> The Overcooked domain uses image-based observations of the entire grid, meaning that all agents receive identical observations. Since there is no partial observability that can be mitigated by  using a centralised critic, we excluded MAPPO from our results for this task. As we mention in the paper, this is consistent with the analysis reported in the original benchmark paper for this domain [1].
>
> > W2. Please provide a discussion on the computation cost compared to other methods.
>
> Compared with other methods, the additional modules we introduce (distance encoder, Fermat state encoder) are only used during the option discovery pre-training phase, which accounts for approximately 5–10\% of the total training time. These modules are removed during downstream task training. Thus, in the computationally demanding online training phase, our method is no more expensive than any standard option-based method, with costs scaling linearly in the number of options. Finally, the model size of these two encoders is relatively small, having the same number of layers as that of a single-agent network.
>
> > W3. Please provide a discussion on the compatibility with mainstream MARL methods, such as. VDN and MAPPO. Can they be integrated together?
>
> In our approach, the discovered joint options are incorporated as additional actions for the learning algorithm; this does not impose any constraints on the choice of MARL model. We opted to use vanilla IQL because it is the simplest method among the baselines, making the benefits of our discovered options easier to isolate. VDN, MAPPO, or any other MARL algorithm can be used too. Please note, however, that temporally extended actions may require different treatments for value-based and policy-based methods, as discussed in prior work, e.g., the Option-Critic framework [2]. Regardless, our method remains fully compatible with any base MARL algorithm, since integrating the discovered options only modifies the action space and does not require architectural changes.
>
> [1] Rutherford et al. JaxMARL: Multi-Agent RL Environments and Algorithms in JAX. NIPS 2024.
>
> [2] Bacon et al. The Option-Critic Architecture. AAAI 2017.

---

### Official Review · Reviewer_TBjq · 2025-10-27

**Soundness:** 3
**Presentation:** 2
**Contribution:** 3
**Rating:** 6
**Confidence:** 3

**Summary:**

This work tackles the problem of option discovery in multi-agent reinforcement learning (MARL), where existing methods (like Eigenoptions) are impractical because they discover too many options, scaling with the massive joint state space. The authors introduce a novel inter-agent relative state abstraction that compresses this joint state space into a compact latent representation. This abstraction is centered around the "Fermat state," which represents the point of maximal alignment among agents, thereby focusing the discovery process on inter-agent relational dynamics rather than the full state space. By discovering options within this compressed, relation-focused space, the method drastically reduces the number of options and encourages the emergence of highly coordinated joint behaviors.

**Strengths:**

1. Novel State Abstraction: The core contribution is a new inter-agent relative state abstraction centered around the "Fermat state." This is a new technique for compressing the massive state space. A direct, practical benefit of this abstraction is that it drastically reduces the number of options that need to be discovered, making the problem computationally tractable. The abstraction is not just a random compression; it's designed to focus the discovery process on inter-agent relational dynamics and synchronization. This is highly relevant for cooperative tasks.

2. The method is empirically shown to discover options that represent highly coordinated joint behaviors, which is the goal of many MARL tasks. The paper demonstrates that its method discovers a more diverse and generalizable set of coordination skills that can better support agents in various downstream tasks compared to other methods.

**Weaknesses:**

1. The authors define the distances between agents' states using the Fermat inter-agent state distance and further approximate this distance by learning a parameterized function. The training process for this function seems to map the joint state space to a single-agent state space by minimizing a temporal distance. The authors claim that this distance metric is capable of comparing the similarity between the states of two agents. However, it is unclear whether this temporal distance can distinguish between the states of two agents, since agents may have different states at the same time step. Moreover, what are the differences between an agent's observations and its 'single state'?

2. The proposed method requires that the MI between each feature distance and the remaining state-feature pairs does not exceed the MI between the feature pairs themselves. Why not directly maximize the MI objective between the state-feature pair and the random vector $Z$? This would also seem to distinguish different features.

3. More evaluations on some commonly used benchmarks, such as SMAC and GRF, should be included.

**Questions:**

Please see the Weaknesses above.

---

> ### Author Response · Authors · 2025-11-25
>
> We thank the reviewer for their feedback. We are pleased that they found our proposed option discovery method "highly relevant" for cooperative scenarios and that they appreciated our empirical results.
>
> > W1. ..., it is unclear whether this temporal distance can distinguish between the states of two agents, since agents may have different states at the same time step. Moreover, what are the differences between an agent's observations and its 'single state'?
>
> Temporal distances are optimized to compare two states, from the same time step, by estimating the number of actions required by a policy to transition from one state (source) to the other (goal). In the homogeneous state setting, we train this module using contrastive learning on states belonging to the same agent’s trajectory; given a source state, we sample goal states from a skewed distribution of future states along that trajectory, akin to the original paper. Closer states are sampled more frequently, enabling the model to learn meaningful temporal distances. For the heterogeneous-state setting, we provide a new analysis and concrete guidelines for extending our method to support such comparisons, as described in Appendix A.8 of the revised paper.
>
> Regarding the reviewer’s second question, there is a subtle distinction between observations and single-agent states. We introduce individual agent states to ensure that absolute-valued information is used during eigenoption discovery. In the pre-training phase, eigenoption methods construct a state–transition graph from random interactions with the environment. In partially observable environments, however, many absolute values are hidden from the agent’s observations, which makes it difficult to reliably distinguish between states. While accessing the underlying state does introduce an additional requirement, our approach (like other methods based on eigenoptions) only relies on this during the short pre-training phase.
>
> > W2. The proposed method requires that the MI between each feature distance and the remaining state-feature pairs does not exceed the MI between the feature pairs themselves. Why not directly maximize the MI objective between the state-feature pair and the random vector? This would also seem to distinguish different features.
>
> If we consider the MI: I($S^{i,j}_f, Z^{i,j}_f$), such an objective would ensure that each feature of $Z^{i,j}$ contains information about the distance associated with its corresponding state feature. However, this alone does not prevent a degenerate solution in which all features of $Z^{i,j}$ encode the same information, which still satisfies this objective. In contrast, our proposed objective explicitly guards against this degenerate case by limiting the information flow from other features into the distance estimate of the current one.
>
> > W3. More evaluations on some commonly used benchmarks, such as SMAC and GRF, should be included.
>
> In light of this and other suggestions, we have included a new experimental setting addressing a limitation of our approach: handling heterogeneous agent states. Specifically, we constructed two additional toy heterogeneous scenarios in which we evaluated the learned inter-agent relative representations, the eigenvectors, and the relative option policies, and provided additional visualisations. Although these settings are based on an adaptation of one of our existing domains (LBF), they do provide a stronger stress test for our relative option-discovery method.

---

### Official Review · Reviewer_gMWz · 2025-10-27

**Soundness:** 3
**Presentation:** 2
**Contribution:** 2
**Rating:** 4
**Confidence:** 3

**Summary:**

This paper tackles multi-agent option discovery by proposing an inter-agent relative state abstraction that centers the joint state around a “Fermat state” of maximal alignment, measures team “spreadness” per feature dimension via learned temporal distances, then discovers options by estimating Laplacian eigenvectors on this relative representation rather than on the raw joint state. The key idea is that synchronisation patterns between agents are a natural substrate for coordination, so compressing the joint state into a feature-wise, alignment-centric embedding both reduces option count and biases discovery toward genuinely joint behaviours. The method trains a Fermat encoder together with a temporal distance model, adds a conditional mutual information regularizer to keep each feature’s distance prediction focused on its own signal, and adapts MacDec-POMDPs to execute joint options under simple team-level synchronisation assumptions. Experiments on Level-Based Foraging and Overcooked show that options learned on the relative representation improve IQL downstream performance and often beat options produced by raw joint state or Kronecker-product constructions.

**Strengths:**

The paper isolates a real bottleneck in multi-agent option discovery, the exponential blow-up of joint state spaces and the tendency to produce loosely coupled skills, then proposes an elegant fix, compress the joint state via a Fermat-state-centred, feature-wise temporal distance representation that foregrounds coordination structure. The learning objective for the Fermat encoder and temporal distances is straightforward, the use of successor-style temporal distances fits the coordination intuition and is representation agnostic, the conditional mutual information penalty is a thoughtful touch to keep features disentangled and to avoid degenerate solutions, the adaptation of MacDec-POMDPs to joint options with simple voting and synchronisation assumptions is clearly described and easy to implement. On the empirical side, the method improves IQL on LBF and Overcooked, it outperforms Kronecker-product options and raw joint-state options, the multi-dimensional variant is especially helpful in the richer Overcooked feature space, and the ablations align with the story the paper tells.

**Weaknesses:**

The empirical scope is still moderate for ICLR standards, two domains and a few scenarios, the evaluation would benefit from additional environments with heterogeneous agent state spaces where the homogeneity assumption breaks, and from tasks with explicit communication limits to stress the information sharing assumption used during joint option selection. Sensitivity to key choices is only partially explored, for example the number of eigenvectors retained, the dataset size for pretraining the distance and Laplacian estimators, the effect of temporal distance quality on option quality, and the robustness of the CMI regularizer. The assumptions for joint option initiation and information sharing are reasonable for a first pass, yet the paper could better characterise failure modes when the “enough agents agree” condition is not met, or when shared observations are stale. Finally, while Figure 2 is very helpful, a more systematic probe into what kinds of synchronisation patterns each discovered option encodes would make the contribution crisper, for example, controlled grid tests that manipulate only one relational factor at a time.

**Questions:**

Can you report sensitivity to the number of eigenvectors and to the size and policy used to collect the 500k transition dataset for distance and Laplacian estimation, in particular, how do option quality and downstream performance vary when the dataset is smaller or when trajectories come from a partially coordinated policy rather than random, can you quantify how the accuracy of the temporal distance model affects discovered options, for example by adding noise or using a weaker successor-distance estimator, can you provide a diagnostic of the synchronisation motifs your eigenoptions encode, for instance by plotting controlled rollouts that show which feature subsets a given option aligns, can you stress test the joint option initiation and information sharing assumptions, for example by injecting delays or dropouts into the shared signals and reporting how often options fail to trigger, can you add a small heterogeneous-agents study, even synthetic, to illustrate how the approach might extend beyond identical S* spaces.

---

> ### Author Response · Authors · 2025-11-25
>
> We are grateful for the reviewer’s assessment of our work. We are pleased that the reviewer recognizes the relevance of our contributions, considers our method as an "elegant fix" to the multi-agent option discovery problem, and identifies the feature disentanglement of the n-distance estimation as a thoughtful addition.
>
> > Q1. Can you report sensitivity to the number of eigenvectors and to the size and policy used to collect the 500k transition dataset for distance and Laplacian estimation, in particular, how do option quality and downstream performance vary when the dataset is smaller or when trajectories come from a partially coordinated policy rather than random?
>
> Approximating graph Laplacian eigenvectors is a well-known and challenging problem, which has received considerable attention in the existing literature ([1, 2, 3]). In our implementation, we adopted one of the most recent approximators, ALLO [1], and aimed to keep the hyperparameters and dataset size as close as possible to those used in the original implementation (200k samples). The decision to increase the dataset size to 500k was motivated by the sensitivity of eigenvector approximators to the size of the state space, which, as mentioned before, grows exponentially with the number of agents. We further clarify that the choice of the number of samples was also based on an empirical assessment of the eigenvector quality obtained by the raw joint-state baseline; we then maintained this number to ensure a fair comparison with our method. Please note that our relative representations provide an abstraction of the joint state space, which in principle should help accelerate the training of such approximators. Nevertheless, the goal of our approach is not to improve the performance of ALLO (or similar methods), but to recover highly coordinated joint behaviours without destabilizing the original option-discovery process.
>
> Regarding sensitivity to the number of eigenvectors, this is primarily an ALLO-specific concern, but we have noticed ALLO to be quite robust (in our experiments) to this hyperparameter. To address how the number of relative eigenvectors (or options) affect downstream performance, we have updated the manuscript to include Figure 5, which quantifies this impact in our experimental domains (Level-Based Foraging, Overcooked), along with a discussion of the results at the end of Section 4. This form of analysis is common in prior option-discovery work, and we thank the reviewer for encouraging us to explore it.
>
> The reviewer’s suggestion to consider “partially coordinated policies" for dataset creation is an interesting research direction. While this is not a standard approach in eigenoption discovery work, the broader question of how to obtain exploration policies that generate higher-quality roll-outs for eigenvector training is indeed compelling, and we believe it represents a promising avenue for further research.
>
> > Q2. can you quantify how the accuracy of the temporal distance model affects discovered options, for example by adding noise or using a weaker successor-distance estimator
>
> While the pair-wise state distance measure is not our primary contribution, the reviewer raises a good point regarding its influence on the n-distance estimation and the overall set of options generated. We motivate our choice in the Related Work section (Section 5), where we argue that in RL environments, temporal distances are the most representative, as they inherently capture environmental dynamics. The quality of the distance estimator directly affects the discovered options, with the resulting synchronisation behaviours aligning agents based on the values of this measure. While this can introduce additional challenges, it also allows us to integrate inductive biases into the type of alignment behaviours obtained. For example, different features can be isolated, excluded from synchronisation considerations, or weighted differently simply by modifying the distance measure.
>
> > Q3. can you provide a diagnostic of the synchronisation motifs your eigenoptions encode, for instance by plotting controlled rollouts that show which feature subsets a given option aligns,
>
> We thank the reviewer for this suggestion. In the revised version of the manuscript, we have added option-policy roll-outs in the main text (Figure 3), where we visualize agent trajectories up to the termination action, along with the position of the fictitious Fermat state and the corresponding n-distance values for the final (goal) state of each option. We also provide additional roll-outs in Appendix A.7 as examples of the coordination behaviours discovered by our method.
>
> [1] Gomez et al. Proper Laplacian Representation Learning. ICLR 2024
>
> [2] Wang et al. Towards Better Laplacian Representation in Reinforcement Learning with Generalized Graph Drawing. ICML 2021
>
> [3] Wu et al. The Laplacian in RL: Learning representations with efficient approximations. ICLR 2019.

---

> ### Author Response · Authors · 2025-11-25
>
> > Q4. can you stress test the joint option initiation and information sharing assumptions, for example by injecting delays or dropouts into the shared signals and reporting how often options fail to trigger
>
> We thank the reviewer for this interesting question. To truly run a meaningful stress test, we would first need to develop a theoretical framework that formally defines the introduction of joint multi-agent options in Dec-POMDPs, and the impact of the quality of information shared between agents. Although this is an idea we intend to explore, it is beyond the scope of this paper. Our objective in this paper was to highlight the shortcomings of previous frameworks for identifying joint options in multi-agent systems, describe a novel approach for discovering such options reliably and efficiently, and to propose simple adaptations that allow us to demonstrate the use of the discovered joint behaviours.
>
> > Q5. can you add a small heterogeneous-agents study, even synthetic, to illustrate how the approach might extend beyond identical S* spaces
>
> Thank you for making making this suggestion, which has helped strengthen the paper. The revised paper now includes (in Appendix A.8) an extension of the initial framework, which was limited to homogeneous state spaces, to theoretically and practically handle heterogeneous cases. The key underlying idea is to redefine S* to allow the pair-wise temporal distance and Fermat encoder modules to operate with distinct agent states spaces. We also adapted one of the domains used in our experimental evaluation (LBF) to include heterogeneous agents, and empirically demonstrated that the extended framework can handle this scenario. We provide visualisations of both the eigenvectors and the option policies learned for the heterogeneous case in two different configurations in this domain. Please note, however, that this adaptation represents a toy environment, and the introduced agent types are not suited for solving the apple-picking task. Furthermore, we also observed a lack of MARL environments supporting heterogeneous agent state spaces, which is another interesting direction for future work.

---

> > ### Comment · Reviewer_gMWz · 2025-11-27
> >
> > Thank you so much for your answers. It addressed my concerns. I would love to see more interesting extended work in the future of your proposed research direction. I would love to raise my score.

---

### Official Review · Reviewer_Sm3d · 2025-10-31

**Soundness:** 3
**Presentation:** 3
**Contribution:** 3
**Rating:** 8
**Confidence:** 1

**Summary:**

This paper presents a framework for discovering temporally extended actions (options) in MARL settings. The authors propose a method that compresses the joint state space into a latent representation centered around a "Fermat state". This abstraction enables the discovery of coordinated behaviors by focusing on inter-agent relational dynamics rather than raw joint states. Empirical evaluations in Level-Based Foraging and Overcooked domains demonstrate that the proposed method outperforms existing option discovery techniques and baseline MARL algorithms, particularly in tasks requiring strong coordination.

**Strengths:**

By introducing the Fermat state and leveraging n-distance metrics, the authors effectively re-center the representation space to emphasize coordination. The use of multi-dimensional disentangled representations allows for richer behavioral diversity, and the mutual information constraint ensures that each feature contributes meaningfully to the learned options.

The empirical results are thorough, covering multiple domains and scenarios, and the visualizations of eigenvectors provide intuitive insights into the discovered coordination patterns. The integration of the framework into the MacDec-POMDP model and the use of decentralized training further enhance its practical relevance.

**Weaknesses:**

I am not familiar with this area, so hard to understand the motivation. It is expected to show more background and examples about the problem.

The reliance on homogeneity among agent state spaces limits its applicability to more diverse multi-agent systems. The assumption of full observability and shared information among agents, while practical for evaluation, may not hold in real-world scenarios.

The computational overhead introduced by multi-dimensional distance estimation and mutual information regularization might be significant.

**Questions:**

Is the proposed option learning generalizable to different environments?

What is the difficult part of extending beyond 2 agents?

---

> ### Author Response · Authors · 2025-11-25
>
> We thank the reviewer for their feedback. We are pleased that they appreciated our solution and found the empirical evaluation thorough, particularly the eigenvector visualizations, which we consider an important part of this work.
>
> > W1. I am not familiar with this area, so hard to understand the motivation. It is expected to show more background and examples about the problem.
>
> We thank the reviewer for their suggestion. We have added Appendix A.1 in the revised manuscript, which presents two realistic examples that demonstrate the importance of discovering options (i.e., coordination strategies) in real-world scenarios, highlight the challenges general multi-agent systems face when learning such strategies from scratch, and underscore the significance of our method in addressing these challenges.
>
> > W2. The reliance on homogeneity among agent state spaces limits its applicability to more diverse multi-agent systems. The assumption of full observability and shared information among agents, while practical for evaluation, may not hold in real-world scenarios.
>
> The reviewer highlights two important limitations of our approach. We address the concern about state homogeneity by adding a comprehensive heterogeneous-state extension of our framework in Appendix A.8, along with an empirical evaluation in two scenarios. This extension can provably handle inter-agent relations for various agent types, enabling different agents to synchronise  the common features of their distinct state spaces. We provide visualizations of both the eigenvectors and the option policies learned for the heterogeneous case.
>
> Regarding full observability, please note that our method assumes only information sharing channels, not full observability. Observation sharing is used in our experiments simply because studying private information transfer through explicit communication channels lies outside the scope of this work. While it is true that such channels may not always be available in real-world settings, we note that some form of information sharing is generally required for any temporally extended behaviors involving multiple actors. That said, we agree that understanding how decentralized multi-agent systems can reliably agree on joint strategies under limited information sharing is an important direction for future research.
>
> > W3. The computational overhead introduced by multi-dimensional distance estimation and mutual information regularization might be significant.
>
> We appreciate the reviewer’s concern regarding computational cost. Please note that the modules mentioned are only utilized during the offline training phase for the discovery of option policies, which accounts for only approximately 5-10\% of total training period. These two modules are then completely discarded when training on the downstream tasks. Furthermore, their number of learnable parameters is only slightly higher than those of a single-agent network, as the number of layers is the same.
>
> > Q1. Is the proposed option learning generalizable to different environments?
>
> Yes, the current framework requires minimal changes when applied to different environments. The main modification is in the definition of the joint state factorization function, which constructs the individual agent states. Most of the modern MARL environments provide such functionality in their practical implementation, where joint states are actually a concatenation of single agent states in the first place. Once single agent states are extracted, our framework can be applied directly. In addition, state access is only required during the offline training period, and is not necessary during downstream task training where the options discovered through our method can be applied like any other.
>
> > Q2. What is the difficult part of extending beyond 2 agents?
>
> Our proposed Fermat n-distance method is well-suited for larger team sizes, as none of the modules in our option-discovery architecture, aside from the Fermat encoder, depend on the number of agents. Moreover, the Fermat encoder can be easily adapted to accommodate larger teams or teams of varying size by replacing the MLP network with a sequence-prediction architecture such as an RNN or a Transformer. There is thus no inherent limitation preventing the method from scaling beyond two agents. In fact, our experiments already include three and four agent settings, as demonstrated by the eigenvector visualizations in Figure 2 and the option-policy roll-outs in Figure 3 of the updated manuscript.

---

### Meta-Review · Area_Chair_z4mP · 2026-01-07

**Summary:**

The paper proposed a new framework for option discovery in MARL settings, which the reviewers found interesting and elegant and the empirical evaluations sufficiently thorough. None of the reviewers founds any critical flaws in the paper and most of the feedback was focused on clarification questions and suggestions that could further improve the paper, and the overall sentiment was quite positive.

The authors addressed most of the major points raised by the reviewers, including adding expanding Appendix A.1 and A.8, clearly explaining the reasoning for their design decisions, and deferring larger requests beyond the scope of the paper to future work. Given the overall positive sentiment and the addressed concerns, I recommend this paper to be accepted.

My only recommendation to the authors is to consider updating the manuscript such that any points that lacked sufficient level of clarity and lead to questions from the reviewers are properly explained / addressed / emphasized in the paper: e.g., making sure to clarify computational overhead of the method (asked about by Sm3d and KghA) and discuss the sensitivity analysis of the Laplacian estimation (requested by gMWz).

**Reviewer Concerns:**

Addressed concerns:
- Concern about lack of support for heterogeneous-state settings (addressed in Appendix A.8)
- Concern about lack of background and motivation for non-experts in the field (addressed in Appendix A.1)
- Concern about lack of sensitivity analysis of the Laplacian estimation (addressed by clarify the adopted approximation method and referring the the relevant literature; I recommend incorporating these clarifications into the paper)
- Request for better visualization of the information encoded in the eigenoptions (addressed in the paper revision)
- Various other clarification questions

Unaddressed concerns and suggestions:
- Request for stress testing the joint option initiation and information sharing assumptions (requires coming up a proper setup for probing assumptions, deemed beyond the scope)
- More evaluations on some commonly used benchmarks, such as SMAC and GRF (the paper already has substantial evaluations, this request further expands the scope w/o clear benefit)

**Reviewer Scores:**

- Reviewer gMWz indicated that they would raise the score. My guess would be they would raise from 4 to 6 or 7 based on discussion.
- Other reviewer scores are 6, 6, 8. Based on the discussion, the reviewers would've likely maintained their scores or perhaps raised scores of 6 by 1 point or so.

---

### Decision · Program_Chairs · 2026-01-26

Accept (Poster)